# Who enjoys solitude? autonomous functioning (but not introversion) predicts self-determined motivation (but not preference) for solitude

Thuy-vy T. Nguyen[1]*, Netta Weinstein[2], Richard M. Ryan[3]

**1** Durham University, Durham, United Kingdom, **2** University of Reading, Reading, United Kingdom, **3** Australian Catholic University, North Sydney, Australia

* vy.nguyen@durham.ac.uk.

**Data Availability Statement:** All relevant data are within the paper.

**Funding:** The author(s) received no specific funding for this work.

## Abstract

Within the solitude literature, two discrete constructs reflect different perspectives on how time spent alone is motivated. *Self-determined motivation for solitude* reflects wanting time alone to find enjoyment and gain meaningful benefits from it, whereas *preference for solitude* concerns wanting time for oneself over others' company regardless of reasons for why time alone is wanted. We investigated two personality characteristics: *introversion* from Big-Five personality theory and *dispositional autonomy* from self-determination theory. In two diary studies university students completed personality measures and reported about their experiences with time spent alone over a period of seven days. Across both studies, contrary to popular belief that introverts spend time alone because they enjoy it, results showed no evidence that introversion is predictive of either preference or motivation for solitude. Dispositional autonomy–the tendency to regulate from a place of self-congruence, interest, and lack of pressure–consistently predicted self-determined motivation for solitude but was unrelated to preference for solitude. These findings provided evidence supporting the link between valuing time spent alone with individual differences in the capacity to self-regulate in choiceful and authentic way.

## Introduction

There has been a growing interest in understanding why people spend time alone [1–4]. This topic is interesting in no small part because pursuits for time alone have been rendered incompatible with our human nature as social organisms [5]. Yet despite the sociality of people, there are times when people seek time alone because they see it as enjoyable, worthwhile, and valuable. Being motivated to spend time alone for those reasons has been referred to as having *self-determined motivation for solitude* [1, 2, 6]–a concept that stems from the self-determination theory (SDT) literature [7, 8], suggesting that motivation for solitude may be personally endorsed and intrinsically motivated. Evidence is mounting that self-determined motivation for solitude is empirically distinct from merely preferring solitude over social time [1, 9, 10],

**Competing interests:** The authors have declared that no competing interests exist.

but there is little knowledge about what drives people to self-determined motivation for solitude, even when they do not necessarily prefer to be alone. In this paper, we explore the role that personality plays in both motivation and preference for solitude, focusing on two personality characteristics that should predict the two constructs differentially: autonomous orientation, which is likely to drive self-determined motivation, and introversion, likely to drive preference.

## Self-determined motivation for solitude

While the benefits of solitude are not new discoveries, it is still common to assume that people dislike being alone and rarely seek it out. Only recently, researchers begin to study different qualities of motivation for solitude and recognize that people do seek out solitude for self-determined reasons. From the SDT perspective, having high self-determined motivation for being alone represents choosing to spend time alone because solitude offers enjoyment or offers some personal benefits. Researchers who have studied this concept [1, 2, 6] contrasted self-determined motivation for spending time alone with motivations that are rooted in feeling like one is forced or coerced into aloneness through other individuals or external circumstance (not self-determined solitude). For example, a person with self-determined motivation to spend time alone will take time out of their day to embrace the benefits that solitude brings, such as opportunities for relaxation, creativity or freedom.

The distinction between choosing solitude because it is valued and feeling forced to be alone are apparent in both methods of measuring self-determined and not self-determined motivation for solitude from the self-determination theory perspective. One operationalization is through the Motivation for Solitude Scale (MSS-SF) [1], which includes items that pertain to reasons for spending time alone for positive and constructive benefits (e.g., creativity, relaxation, self-discovery) and those that concern reasons for being alone due to not feeling oneself or accepted around other people. The MSS-SF specified different positive and negative reasons for approaching solitude. The other, adapted from the Self-Regulation Questionnaire by Ryan and Connell [11], directly asks whether the reasons for spending time alone have to do with finding it enjoyable or valuable, and also asks whether participants are alone because of their internal compulsion (i.e., feeling like one should be alone) or some external circumstances (i.e., being made to be alone). The latter measure phrases items more broadly so they can be used in laboratory settings or daily assessments where participants may not have specific reasons for why they would like to be alone at that moment or on a specific day.

Both scales have been used to demonstrate that self-determined motivation correlates with positive functioning in general [1], as well as with more proximal outcomes [2]. These positive reasons for spending time alone have been documented in studies not only from Western samples [1, 12], but also in one East Asian sample [13].

## Distinction between preference for solitude and self-determined motivation for solitude

From an SDT perspective, self-determined motivation does not concern with people's behaving in ways that are in line with their preference [14]. In other words, when a person prefers a course of action, this preference might be based on available options rather than reflecting personal interests or values. As such, in relation to spending time alone, self-determined motivation for solitude ought to be differentiated from the *preference* that someone might have at any given time when they decide between two available options: being alone versus being with others. Preference for solitude is a concept rooted in the social approach and social avoidance motivation literature. Within this perspective, preference for solitude represents a motivation

to move away from social interactions to avoid undesired interactional outcomes, and has been reported in young children [4, 15] and adolescents [16–18]. Among adults, preference for solitude has been showed to be associated with trait loneliness [19] and previous experiences of ostracism [20].

Originally, preference for solitude has been studied as a form of social withdrawal in young children. However, it was stressed that, different from other well-studied forms of social withdrawal, preference for solitude is not underlined by shyness or social anxiety [4, 21]. Such discovery gave rise to a new phenomenon referred to as unsociability [4]–a term used to characterize some children's tendency to spend time or play alone, whose behavior is not driven by social difficulties or unpleasant interactions with peers. Other researchers have described adults who display similar patterns of behaviors as those who have high preference for solitude [17, 20] or possess a solitropic orientation [22]. Although these different terms have been used to describe the propensity toward spending time alone over interacting with others (e.g., unsociability, preference for solitude, solitropic orientation), in this paper we will generally refer to this phenomenon as preference for solitude.

To summarize, the two literatures identify disparate outcomes when one is seeking solitude. Self-determined motivation has been consistently operationalized as a form of a healthy and adaptive motivation for spending time alone [1, 6, 9] that yields positive associations with well-being correlates. On the other hand, the way that preference for solitude has been operationalized has led to its associations with maladaptive constructs like loneliness [1, 19, 23] and social anxiety [1]. More importantly, while the literature clearly demonstrated that these two concepts are independent of one another, the correlations between measures of self-determined motivation for solitude and preference for solitude have been inconsistent. Burger's 12-item measure of preference for solitude [19] showed positive correlations with measures that Thomas and Azmitia [1] used to assess both self-determined and not-self-determined motivation for solitude in late adolescence. Another study that examined preference for solitude in adults older than 35 years of age showed a positive correlation between Burger's preference for solitude scale and self-determined motivation for solitude, and also a negative correlation between preference and *not* self-determined motivation for solitude [10]. These correlations suggest that, for those beyond adolescence and young adulthood, preference for solitude is associated more with self-determined rather than not-self-determined motivation for solitude, whereas preference for solitude in late adolescence could reflect both types of motivation.

In this paper, we explore two key personality predictors that may help to explain how both motivation and preference are shaped through the relationship that people have towards solitude, or time spent with themselves. We approach the question of what sets motivation and preference apart from the view that stable dispositions can predispose individuals to a type of relationship with solitude; for example, one characterized by self-determined motivation, or another kind, for example one that is predominantly characterized by preference.

## Dispositional autonomy as predictor of self-determined motivation for solitude

Identifying dispositional characteristics that may predict daily self-determined motivation for solitude can be informed by the developmental psychology literature. First, it is important to highlight that solitude is not inherently comfortable, and developmental psychologists suggest that a healthy attitude toward solitude, despite the challenges of being alone as social animals, signals positive development and emotional maturity [24–26]. Particularly, the prominent developmental psychoanalyst Donald Winnicott, asserted that the capacity to be alone is an ability that develops from a nurturing environment where children are allowed opportunities to express their freedom

through play without intrusion from caregivers. Indeed, this view of healthy motivation for solitude has been reflected in recent empirical works, which showed that the pursuit of solitude for its benefits to creativity and relaxation was linked to experiences of personal growth in young adults [1]. When individuals experience positive solitude, they often attribute this to the freedom to engage in chosen activities and the removal of social pressure and surveillance [12]. From these literatures, our research was aimed at testing the hypothesis that solitude allows opportunities for autonomous self-expression–being able to engage in activities that reflect one's interests and choices. To test this hypothesis, one concept in social-personality psychology that captures the propensity for autonomous self-expression is dispositional autonomy.

*Dispositional autonomy* refers to the degree to which an individual tends to regulate their behaviors in a more or less autonomous, or choiceful, manner–that is, endorsed by the self and motivated through interest or valuing [27]. Building on early SDT theorizing [28], dispositional autonomy is comprised of three components that together make up the concept: 1) a tendency toward self-congruence, 2) an ability to take interest in one's emotional experiences, and 3) a lower vulnerability to momentary pressures and controls [27]. Findings show that individuals with this disposition experience daily activities more positively, and better integrate life experiences into their other views of the self with less internal conflict [29]. Further, dispositional autonomy seems important for felt comfort with one's experiences of the self [30], and for making sense of potentially conflictual emotions [31].

Previous literature has demonstrated that those who behave in ways that are consistent with their beliefs and values tend to be motivated by intrinsic and self-determined reasons in whatever they do [29, 30, 32]. However, it is important to highlight that this might not be true for solitude, which is commonly portrayed as a challenging experience for people. This is particularly relevant for young adults because they find solitude more difficult than older age groups [33, 34] unless they have been allowed opportunities to develop a capacity to enjoy it. Given this, an investigation of dispositional autonomy that represents that capacity speaks to Winnicott's theorizing [25] discussed above, which has not been considered in previous studies. As such, while no empirical findings have linked dispositional autonomy to solitary experiences, we considered dispositional autonomy as an important predictor of the capacity to be alone because of a healthy and non-defensive relationship with oneself [34]. We expected that, because individuals high in dispositional autonomy would be likely to enjoy their inner world as much as other people's company, dispositional autonomy would yield no significant association with preference for solitude over social interactions. Nonetheless, dispositional autonomy would relate more to the enjoyment of solitude for its own sake; that is, a person who is high in dispositional autonomy would show higher levels of self-determined motivation for solitude.

## Introversion as a predictor of preference for solitude

More widely studied than dispositional autonomy, when it comes to predicting who likes spending time alone, laypersons and researchers alike intuit that *introversion* must play an important role. An introvert is someone who is reserved and inhibited, so it is reasonable that they would prefer being alone to being in social interactions [e.g., 35–37]. It has been showed that individual difference in preference for solitude correlate positively with introversion [19]. Another study by Leary, Herbst, and McCrary [22] that collected data from a variety of different personality measures showed that several traits related to the introversion-extraversion dimension of personality, such as sociability, extraversion as measured by the Big-Five inventory, need to belong, and desire for social contact, correlated negatively with how frequently participants reported engaging in solitary activities. Similar patterns of findings were also shown in relation to the likelihood of people reporting they would do certain activities alone.

The authors also found that these characteristics that pertain to being a sociable, extraverted, and people-oriented person also correlated with deriving less enjoyment from engaging in activities alone. These findings were taken to mean that those who lack those qualities of being sociable, extraverted, and people-oriented should prefer more aloneness and enjoy it more.

However, while Burger [19] and Leary et al. [22] used one-time measures of people's evaluations of their own personality and attitudes toward solitude, a study by Srivastava et al. [38], which used a day-construction design and collected data from participants' day-to-day experiences, showed a different pattern. When asked to report levels of positive affect while interacting with other people and while not interacting, those high in extraversion reported feeling more positive in relation to both types of experiences. Interestingly, those low in extraversion also felt more positive in social interaction than when not interacting, and in fact felt less positive than extraverts when not interacting with others. This presents an interesting picture for introverts when it comes to their time spent alone; they generally show a greater preference for solitude than extraverts [1], because they do not derive as much benefit out of social interactions as do extraverts, but they might not necessarily enjoy time alone more [39]. In other words, introverts' preference for solitude might be driven more by the lack of appeal held by available social experiences, and less by their anticipation that spending time alone would be enjoyable. As such, and consistent with the recent findings by Thomas and Azmitia [1], we predicted that scores on introversion would positively relate to ratings of daily *preference for solitude*. In this research, we operationalized introversion as having the opposite qualities to those that are associated with extraversion; that is, introverts tend to be more reserved, quieter, less talkative, energetic, and assertive. We will describe how we assess introversion in the method section. On the other hand, we did not expect that introversion would positively relate with *self-determined motivation for solitude* because someone who possesses those qualities described below might prefer to have more time for themselves, but they might do so for either self-determined or not self-determined reasons.

## The present research

This research was conducted to differentiate preference for solitude and self-determined motivation for solitude through understanding their different links to personality characteristics. Informed by theorizing and extant research discussed above, we predicted that introversion would only relate to preference for solitude but not necessarily to self-determined motivation for solitude. On the other hands, we predicted dispositional autonomy–the capacity to self-regulate in an autonomous and volitional way—would relate to self-determined motivation for solitude but not to preference for solitude. Thus, the present empirical tests served to establish discriminant validity of the two constructs by investigating their personality underpinnings.

To do so we conducted two daily diary studies. The advantage of this design is that it allows us to capture how personality characteristics relate to solitary experience on a daily basis, an advance compared to previous methodologies that relied on retrospective self-reports collected at one time of people's experiences with being alone. This approach also had some advantages relative to lab experiments on solitude [2] in that it allowed a window into experiences as they naturally occur in people's' daily lives. Specifically, we tested two confirmatory hypotheses:

1. Dispositional autonomy would show positive association with self-determined motivation for solitude. We did not predict that dispositional autonomy would correlate with preference for solitude.

2. Introversion would show positive association with preference for solitude. We did not predict that introversion would correlate with self-determined motivation for solitude.

## Study 1

### Study 1 method

**Participants.** One hundred and eighty-three undergraduate students (153 women) between the age of 18 and 28 years ($M_{age}$ = 20 years, $SD$ = 1.36) participated in the study, after excluding five who signed up but either did not complete the initial survey (three participants) or did not participate in the diary portion of the study (two participants). Sample sizes were not determined based on power analyses. Rather, we determined sample size based on a realistic expectation of how many participants we could recruit from the pool that was available. We selected to test study hypotheses with students because we could access and track this population, and because there was no external funding available to recruit community participants outside of undergraduate participant pool.

However, it is worth noting that this age group of emerging adults has been the focus of past research that has shown consistent links between self-determined motivation and well-being in solitude [1, 2], adding value to our current tests of antecedents of self-determined motivation. The final sample provided 1227 units (in days) of data. The sample consisted of 53% Whites and Caucasian participants, 35% Asians or Pacific Islander participants, 5% Black or African American participants, and 7% participants of other races or multi-races. Eighty percent of participants completed all seven days of diary, 14% completed 6 days, 3% completed 5 days, less than 2% completed 4 days and less than 2% completed 3 days.

**Procedure.** Two-hundred time slots were posted for a duration of one week. Participants were instructed they would complete an initial survey to enroll into the email list, and that they will receive daily surveys starting on Monday of the following week. In all, participants were provided with eight surveys: One initial survey that assessed their personality, and seven daily diary surveys. Prior to the diary surveys, participants completed a questionnaire that included questions about their personality. Subsequently, daily surveys were sent by email after 5PM each day and these surveys were set to expire at 6AM the next day to prevent delayed responding. Each survey took approximately 20 to 30 minutes to complete. Each survey completed was incentivized with extra course credit, and those who completed all eight surveys were enrolled into a lottery drawing for a $25 gift card.

*Initial survey*. All participants were enrolled after submitting an initial survey. Included in the initial survey were several personality measures, including the Introversion subscale from the Big-Five Inventory [40] and the Index of Autonomous Functioning (IAF) [27]. Descriptions of personality measures used in this study are provided below.

*Dispositional autonomy*. Dispositional autonomy was measured with the 15-item Index of Autonomous Functioning [27]. This measure captures three aspects of an autonomous disposition, namely, a tendency to act in congruence with one's values and beliefs, a reliance to being susceptible external controls and pressures, and a tendency to take an interest in one's internal feelings and experiences. Participants rated their responses on a 5-point Likert-scale, ranging from 1 = "*Strongly disagree*" to 5 = "*Strongly agree*". Items include "My decisions represent my most important values and feelings", "I do things in order to avoid feeling ashamed", and "I am deeply curious when I react with fear and anxiety". All fifteen items were averaged after items from the susceptibility to pressure subscale were reversed, as recommended by Weinstein et al. [27]. Overall scale reliability was acceptable ($\alpha$ = .78).

*Introversion*. To measure introversion, we used the eight items from John and Srivastava's [40] Big-Five Inventory (BFI) that were used to measure the Big-Five trait of extraversion. This measure includes descriptive statements of how extraverts often behave, such as being "talkative" and "full of energy". We used this instead of the NEO-PI (Neuroticism, Extraversion, Openness Personality Inventory [41]) because the NEO-PI includes positive emotions and

warmth as descriptors of extraversion, which could potentially portray introversion as lacking those positive qualities (see comparison between NEO-PI and BFI in Zillig et al [42]). Using the BFI Extraversion subscale, we showed participants a stem stating, "I see myself as someone who…", and they proceeded to rate their agreement to a series of descriptive statements. We reverse-coded the extraversion-related items like "talkative" and "full of energy", and averaged them with items included "reserved" and "quiet", such that higher overall scores reflected introversion ($\alpha$ = .87). Participants rated their responses on a 5-point Likert scale, ranging from 1 = "*strongly disagree*" to 5 = "*strongly agree*".

*Diary survey*. Each day, participants received an email with links to diary surveys that are made available every day at 5PM and expired at 6AM. Participants were instructed to complete the survey at any times between this interval that were convenient to them. For each diary survey, participants were asked about a significant event that happened to them that day, their experiences of such event (e.g., autonomy, need satisfaction, positive and negative affect), their self-determined motivation for solitude and levels of preference for solitude. Only self-determined motivation for solitude and preference for solitude were the variables of interest for this present paper, whereas other questions were part of a separate project studying the link between daily positive events and later memories of such events. Because our survey could not capture momentary experiences with solitude (an experience sampling design would be more appropriate), we did not ask participants about any specific solitary experiences but only asked about their general evaluations of their daily experiences with it.

*Self-determined motivation for solitude*. For this variable, we used the scale from Nguyen et al. [2], which we felt was more appropriate for a diary study design because it measures state-level motivation for solitude and could better capture day-to-day fluctuation. In comparison, the measure developed by Thomas and Azmitia [1] is more appropriate for distinguishing individual differences in self-determined motivation for solitude. For our measure, respondents were presented with a prompt: "Different people spend time by themselves for different reasons. Please indicate the extent to which each of the following reasons applies to you regarding all the instances when you were by yourself today". Four items assessed self-determined motivation for solitude, namely the reasons for spending time alone because of the value in, and enjoyment of, the activity. Respondents indicated their agreement on scales from 1 (*not at all true*) to 7 (*very true*). Those items were: "I simply was enjoying my time alone for its own sake", "I was alone because having time to myself is an important part of my day", "I was alone because solitude is one of the things I value in my life", and "I found it enjoyable to be in my own company". Items ($\alpha$ = .91) yielded high reliability across all items and assessments. Therefore, we averaged all items for a total score reflecting higher self-determined motivation for solitude reported on each of the seven days.

*Preference for solitude*. We adapted items from a measure by Wang et al. [18], which assessed preference for solitude—the degree to which participants desired to be alone more than with others. We decided to use this measure rather than Burger's preference for solitude [19] because Wang et al.'s measure is shorter and thus less burdensome to participants who complete it repeatedly throughout the daily diary procedure. Further, Burger's measure includes certain items that describe specific situations which might not apply to participants' daily experiences, such as "I like to vacation in places where there are few people around and a lot of serenity and quiet" or "If I were to take a several-hour plane trip, I would like to sit next to someone who was pleasant to talk with". Some of Burger's items also conflate preference and desire to be alone with enjoyment and benefits of being alone (i.e., "I enjoy being by myself", Time spent alone is often productive for me"). Because we aimed to distinguish preference and desire from motivation to be alone for the enjoyment and benefits of solitude, we opted not to include those items in our measure. Instead, the items we used were "Today I

wanted to be by myself rather than with others", "Today, I would prefer being with other people than being by myself" (reverse coded), and "Today, I had a strong desire to get away from others to be by myself". We modified Wang et al.'s original measure, which they used for children between 8th and 12th grades, so that the items are more appropriate for the ages of our sample. Participants indicated their agreement on scales from 1 (*not at all true*) to 7 (*very true*). These items were averaged to create a composite score for daily preference for solitude. Items yielded high reliability across all items and assessments ($\alpha = .99$).

*Ethics statement.* Both studies were reviewed and approved by the Research Subjects Review Board at the University of Rochester (Study 1: RSRB00070882; Study 2: RSRB3612) before the studies began. For both studies, participants were provided an information letter hosted online. The participants were asked to confirm their consent to participation by clicking a "Next" button to proceed to the initial survey to report their demographic information and fill out personality questionnaire. The information letter specifically instructed the participants that they can skip any of the questions that they did not want to answer or withdraw their participation at any time without penalty. The same consent form was presented in all phases of the study.

*Analytic strategy.* Random-intercept models were conducted using the 'lme4' package in R program using full maximum likelihood for parameter estimations of both fixed and random effects. All data and code are shared on OSF (link: https://osf.io/yfgnm/?view_only=fcb98c41581c49ef9c2fd8c6765d6467).

## Study 1 results

**Scale reliability.** Analyses of reliability for preference for solitude items revealed small proportion of variance explained by between-person differences while there was more person-by-day variation, suggesting that preference for solitude varies differently on a day-to-day basis depending on individuals. On the other hand, we observed large proportion of variance explained by between-person differences for self-determined motivation for solitude, suggesting that we had some participants that were higher or lower in general on this variable across all days and all items. We also observed person-by-day variation, meaning daily variation on this measure differed across individuals as well. Nonetheless, for both measures, there was not much variance at the day-by-item or person-by-item levels, suggesting that we had high reliability for items across days and across participants (.99 for preference and .91 for self-determined motivation) and items could be combined in composites for each day.

**Confirmatory analyses.** The daily average for self-determined motivation for solitude was $M = 4.37$ (*Median* = 4.5, $SD = 1.81$), and for preference for solitude was $M = 3.51$ (*Median* = 3.33, $SD = 1.59$). Both introversion and dispositional autonomy were entered simultaneously into the random-intercept regression models, one defining preference for solitude as outcome and the other defining self-determined motivation for solitude as the outcome. Both models satisfied normality assumption and revealed an ICC of .56 for self-determined motivation and .23 for preference for solitude.

*Self-determined motivation for solitude.* In the model predicting self-determined motivation for solitude (Table 1), results indicated a positive association between dispositional autonomy and self-determined motivation for solitude ($\beta = .14$, CI 95% = [.03, .26], $t(1221) = 2.40$, $p = .016$). Controlling for gender and ethnicity revealed similar results ($\beta = .18$, CI 95% = [.06, .29], $t(1221) = 2.92$, $p = .004$). Therefore, our second hypothesis was supported. We did not find significant association between introversion and self-determined motivation for solitude ($\beta = .07$, CI 95% = [-.04, .19], $t(1221) = 1.23$, $p = .220$).

*Preference for solitude.* Results (Table 2) did not show a significant association between introversion and preference for solitude ($\beta = .03$, CI 95% = [-.06, .12], $t(1221) = 0.72$, $p = 0.470$). Therefore,

**Table 1. Random-intercept models that include introversion and dispositional autonomy predicting daily self-determined motivation for solitude (Study 1).**

| Predictors | Self-determined motivation | | | | | | Self-determined motivation (controlling for gender & ethnicity) | | | | | |
|---|---|---|---|---|---|---|---|---|---|---|---|---|
| | B | CI(B) | ß | CI(ß) | t | p | B | CI(B) | ß | CI(ß) | t | p |
| (Intercept) | 2.05 | [.16, 3.94] | .00 | [-.11, .12] | 2.13 | **.034** | 1.02 | [-1.01, 3.04] | -.31 | [-.65, .04] | .99 | .325 |
| Introversion | .14 | [-.11, .39] | .07 | [-.05, .18] | 1.11 | .266 | .15 | [-.09, .40] | .07 | [-.04, .19] | 1.23 | .220 |
| Dispositional autonomy | .53 | [.10, .97] | .14 | [.03, .26] | 2.40 | **.016** | .65 | [.21, 1.09] | .18 | [.06, .29] | 2.92 | **.004** |
| Gender [woman vs. others] | | | | | | | .37 | [-.16, .90] | .20 | [-.09, .50] | 1.35 | .176 |
| Ethnicity [Asian vs. others] | | | | | | | .64 | [.04, 1.24] | .35 | [.02, .69] | 2.10 | **.036** |
| Ethnicity [White vs. others] | | | | | | | .10 | [-.46, .66] | .06 | [-.25, .37] | 0.35 | .724 |
| **Reliability estimates and estimates of between-person reliability and reliability of change** | | | | | | | | | | | | |
| | Variance | Percent | | | | | | | | | | |
| $\sigma^2_{Person}$ | 1.76 | .47 | | | | | | | | | | |
| $\sigma^2_{Day}$ | -0.01 | .00 | | | | | | | | | | |
| $\sigma^2_{Item}$ | 0.03 | .01 | | | | | | | | | | |
| $\sigma^2_{Person \times Day}$ | 1.26 | .33 | | | | | | | | | | |
| $\sigma^2_{Person \times Item}$ | 0.10 | .03 | | | | | | | | | | |
| $\sigma^2_{Day \times Item}$ | 0.01 | .00 | | | | | | | | | | |
| $\sigma^2_{Error}$ | 0.62 | .16 | | | | | | | | | | |

our first hypothesis was not supported. We also did not find significant association between dispositional autonomy and preference for solitude (ß = -.01, CI 95% = [-.10, .08], $t(1221)$ = -0.23, $p$ = 0.821). Controlling for gender and ethnicity did not change the results of this model.

## Study 2

In Study 1, we found that dispositional autonomy predicted greater self-determined motivation for solitude. We did not find evidence for the expected association between introversion

**Table 2. Random-intercept models that include introversion and dispositional autonomy predicting daily preference for solitude (Study 1).**

| Predictors | Preference for solitude | | | | | | Preference for solitude (controlling for gender & ethnicity) | | | | | |
|---|---|---|---|---|---|---|---|---|---|---|---|---|
| | B | CI(B) | ß | CI(ß) | t | p | B | CI(B) | ß | CI(ß) | t | p |
| (Intercept) | 3.46 | [2.20, 4.72] | .00 | [-.08, .09] | 5.40 | < .001 | 2.98 | [1.61, 4.35] | -.18 | [-.44, .08] | 4.28 | < .001 |
| Introversion | .06 | [-.10, .23] | .03 | [-.06, .12] | 0.72 | .470 | .07 | [-.09, .24] | .04 | [-.05, .13] | 0.88 | .379 |
| Dispositional autonomy | -.03 | [-.32, .26] | -.01 | [-.10, .08] | -0.23 | .821 | .01 | [-.29, .31] | .00 | [-.09, .09] | 0.07 | .945 |
| Gender [woman vs. others] | | | | | | | .18 | [-.18, .54] | .11 | [-.11, .34] | 0.99 | .323 |
| Ethnicity [Asian vs. others] | | | | | | | .27 | [-.14, .67] | .17 | [-.09, .42] | 1.30 | .194 |
| Ethnicity [White vs. others] | | | | | | | .10 | [-.28, .48] | .06 | [-.17, .30] | 0.52 | .606 |
| **Reliability estimates and estimates of between-person reliability and reliability of change** | | | | | | | | | | | | |
| | Variance | Percent | | | | | | | | | | |
| $\sigma^2_{Person}$ | 0.44 | .11 | | | | | | | | | | |
| $\sigma^2_{Day}$ | 0.05 | .01 | | | | | | | | | | |
| $\sigma^2_{Item}$ | 0.25 | .07 | | | | | | | | | | |
| $\sigma^2_{Person \times Day}$ | 1.46 | 0.38 | | | | | | | | | | |
| $\sigma^2_{Person \times Item}$ | 0.39 | .10 | | | | | | | | | | |
| $\sigma^2_{Day \times Item}$ | 0.00 | .00 | | | | | | | | | | |
| $\sigma^2_{Error}$ | 1.23 | .32 | | | | | | | | | | |

and preference for solitude. It is important to note that preference for solitude in Study 1 appeared to vary highly on a day-to-day basis within individuals, suggesting that variation in preference for solitude might be less a function of individual differences and more a function of experiences within the day. However, this finding could have resulted from a lack of between-person variation in this particular sample, so we recruited a larger sample in Study 2 and tested the same hypotheses again. Additionally, because we only compared predictive ability of dispositional autonomy to introversion, we couldn't draw a conclusion on whether dispositional autonomy represents a unique individual difference that predicted self-determined motivation for solitude above and beyond the other Big-Five traits. Particularly, dispositional autonomy shared some variance and positive correlations with openness to experience, agreeableness, and conscientiousness dimensions of Big-Five personality [43]. A meta-analysis of experience-sampling studies also showed that all Big-Five traits explained significantly individuals' daily state-level experiences and behaviors [44]. So, there is a case to be made about the ability of dispositional autonomy to predict self-determined motivation for solitude above and beyond Big-Five personality traits. As such, in Study 2, the same design was adopted with two improvements: 1) We used a larger sample size, and 2) We included all Big-Five traits to our mixed-effect models.

## Study 2 method

**Participants.**   Three hundred sign-ups were made available to undergraduate students, and 287 participants between 18 and 28 years of age (186 women, 99 men, 1 unspecified) completed the initial survey. Similar to Study 1, sample size was not determined by power analysis but based on realistic expectation of how many participants we can recruit from the available pool. The sample was predominantly White (43%) and Asian (42%) participants, while Black participants made up 9% and Hispanic or Latino participants made up approximately 10% of the sample.

**Procedure.**   When participants signed up, they were asked to complete a questionnaire consisting of the dispositional autonomy ($\alpha$ = .76) and Big-Five introversion ($\alpha$ = .87) scales. Study 2's design was the same as Study 1, except for the addition of all other Big-Five traits. Specifically, this time we evaluated the extent to which introversion and dispositional autonomy each predicted solitude-relevant outcomes above and beyond agreeableness ($\alpha$ = .79), conscientiousness ($\alpha$ = .78), neuroticism ($\alpha$ = .81), and openness to experience ($\alpha$ = .72). All Big-Five traits, including the subscale used in Study 1 to measure introversion, were assessed using the Big-Five Inventory (40). Again, participants were presented with a stem: "I see myself as someone who. . .". Agreeableness was measured with items such as "is helpful and unselfish with others". Conscientiousness was measured with items such as "does a thorough job". Neuroticism was measured with items such as "worries a lot". Openness to experience was measured with items such as "values artistic, aesthetic experiences". All items are included in codebooks shared on our OSF folder.

After participants filled out the initial questionnaire, on Monday of the week following signing up, participants completed daily surveys for five days until Friday. For the diary surveys, participants reported their preference for solitude ($\alpha$ = .89) and self-determined motivation for solitude ($\alpha$ = .97) on that day. Participants also reported on other experiences such as their social media usage and the extent to which they experienced their social interactions and alone time as being authentic or inauthentic; those data were collected for another project. Only the data on preference for solitude and self-determined motivation for solitude are described here. Sixteen participants who filled out the initial survey did not the diary portion of the study and one participant had missing values on the personality data; those were dropped from the data set. The final sample has 270 participants who provided a total of 1150 units (in days) of data.

*Analytic strategy*. Similar to Study 1, we conducted two separate random-effects regression models using maximal likelihood estimations. The models included Big-Five personality traits and dispositional autonomy as simultaneous predictors on preference for solitude and self-determined motivation for solitude.

## Study 2 results

**Scale reliability.** Once again, we observed that preference for solitude appears to vary on a day-to-day basis compared to self-determined motivation for solitude, and that the larger percentage of variance in self-determined motivation for solitude could be accounted for by individual differences (see variance components in Tables 3 and 4). Nonetheless, for both measures, there was not much variance at the day-by-item or person-by-item levels, suggesting that we had high reliability for items across days and across participants (.89 for preference and .97 for self-determined motivation) and items could be combined in composites for each day within each participant.

**Confirmatory analyses.** The daily average for self-determined motivation for solitude was $M = 4.43$ (*Median* = 4.5, $SD = 1.63$) and for preference for solitude was $M = 3.85$ (*Median* = 4, $SD = 1.39$). Both introversion and dispositional autonomy, and all other Big-Five traits, were entered simultaneously into the random-intercept regression models, one defining preference for solitude as outcome and the other defining self-determined motivation for solitude as the outcome. Both models satisfied normality assumption and revealed an ICC of .44 for self-determined motivation and .29 for preference for solitude.

The model predicting self-determined motivation for solitude showed that dispositional autonomy was a significant predictor ($\beta = .11$, CI 95% = [.00, .21], $t(1140) = 1.97$, $p = .049$). Additionally, introversion yielded significant positive association in this model ($\beta = .10$, CI 95% = [.01, .20], $t(1140) = 2.14$, $p = .033$), and conscientiousness also yielded significant positive association with self-determined motivation for solitude ($\beta = .10$, CI 95% = [.00, .54], $t(1140) = 2.00$, $p = .046$), but other Big-Five traits did not (see *Table 3*). However, once gender and ethnicity were controlled for in the model, all three predictors, including dispositional autonomy, introversion, and conscientiousness, no longer predicted self-determined motivation for solitude. In this latter model with covariates, being a woman was associated on average with greater self-determined motivation for solitude ($\beta = .26$, CI 95% = [.06, .46], $t(1137) = 2.50$, $p = .013$).

The models (Table 4) predicting preference for solitude did not show evidence of significant association between introversion and preference for solitude ($\beta = .080$, CI 95% = [-.01, .16], $t(1141) = 1.74$, $p = .083$). Again, in Study 2, with a larger sample size, our hypothesis was not supported. However, we found that agreeableness emerged as a significant predictor of preference for solitude, and the association was in the negative direction ($\beta = -.18$, CI 95% = [-.27, -.09], $t(1141) = -3.96$, $p < .001$). This suggested that those who have a friendly and warm personality might be less likely to prefer spending time alone. The results remained similar when we controlled for gender and ethnicity (see *Table 4*).

## General discussion

In two daily diary studies we examined how personality characteristics are linked to individuals' motivation and preference for solitude. This research distinguished between two phenomena in the solitude literature: First, daily self-determined motivation for solitude refers to the motivation for time alone in order to find enjoyment and gain meaningful benefits from it, and second, daily preference for solitude refers to favoring being alone rather than being with other people. We proposed that these two phenomena are conceptually distinct and therefore

**Table 3. Random-intercept models that include introversion, dispositional autonomy, and all other big-five traits, predicting daily self-determined motivation for solitude (Study 2).**

| Predictors | Self-determined motivation | | | | | | Self-determined motivation (controlling for gender & ethnicity) | | | | | |
|---|---|---|---|---|---|---|---|---|---|---|---|---|
| | B | CI(B) | ß | CI(ß) | t | p | B | CI(B) | ß | CI(ß) | t | p |
| (Intercept) | 2.04 | [.08, 4.00] | .01 | [-.08, .09] | 2.05 | **.041** | 2.51 | [.48, 4.54] | -.14 | [-.36, .07] | 2.43 | **.015** |
| Agreeableness | -.25 | [-.52, .02] | -.09 | [-.19, .01] | -1.79 | .074 | -.26 | [-.53, .00] | -.10 | [-.20, .00] | -1.94 | .053 |
| Neuroticism | -.10 | [-.32, .12] | -.04 | [-.14, .05] | -0.87 | .382 | -.19 | [-.42, .05] | -.08 | [-.19, .02] | -1.56 | .119 |
| Openness to experience | .21 | [-.08, .51] | .07 | [-.03, .17] | 1.43 | .154 | .18 | [-.11, .47] | .06 | [-.04, .16] | 1.24 | .214 |
| Conscientiousness | .27 | [.00, .54] | .10 | [.00, .20] | 2.00 | **.046** | .24 | [-.03, .52] | .09 | [-.01, .19] | 1.77 | .077 |
| Introversion | .21 | [.02, .40] | .10 | [.01, .20] | 2.14 | **.033** | .19 | [-.00, .38] | .09 | [-.00, .19] | 1.95 | .052 |
| Dispositional autonomy | .38 | [.00, .76] | .11 | [.00, .21] | 1.97 | **.049** | .35 | [-.03, .73] | .10 | [-.01, .21] | 1.81 | .071 |
| Gender [woman vs. others] | | | | | | | .42 | [.09, .75] | .26 | [.06, .46] | 2.50 | **.013** |
| Ethnicity [Asian vs. others] | | | | | | | .03 | [-.35, .41] | .02 | [-.21, .25] | 0.17 | .865 |
| Ethnicity [White vs. others] | | | | | | | -.12 | [-.50, .26] | -.07 | [-.31, .16] | -0.62 | .532 |

**Reliability estimates and estimates of between-person reliability and reliability of change**

| | Variance | Percent |
|---|---|---|
| $\sigma^2_{Person}$ | 1.02 | .33 |
| $\sigma^2_{Day}$ | 0.01 | .00 |
| $\sigma^2_{Item}$ | 0.02 | .01 |
| $\sigma^2_{Person \times Day}$ | 1.21 | .39 |
| $\sigma^2_{Person \times Item}$ | 0.08 | .02 |
| $\sigma^2_{Day \times Item}$ | 0.00 | .00 |
| $\sigma^2_{Error}$ | 0.73 | .24 |

could be associated with different personality traits. Specifically, we argued that self-determined motivation–pursuing solitude because of the value, enjoyment, and personal importance of it, requires a comfort with and even propensity towards self-regulation and therefore should be predicted by autonomous functioning. On the other hand, no such self-regulation was necessary in order to merely want to be alone more so than spending time with other people.

Our hypotheses were driven by the previous literature suggesting that preference for solitude was generally associated with traits related to disfavoring social interactions due to either previous negative experience (i.e., history of social exclusion [16, 34]) or experiencing low belongingness to one's social groups (i.e., loneliness [15, 18]). As such, we expected that individuals who were high on introversion would be disposed to preferring solitude in their daily lives. On the other hand, while there has not been much research directly investigating personality correlates of self-determined motivation for solitude, previous research suggests that it signifies positive development in emerging adults [1], and better regulatory capacity in adults [2]. We therefore expected that individuals higher in dispositional autonomy would be more disposed to daily self-determined motivation for solitude in their daily lives.

Dispositional autonomy was made up of three components that are thought to move individuals closer to their selves and promote self-driven functioning, including the propensity to behave in accordance to one's belief and values, the ability to not be carried away by social influences and internal negativity, and the tendency to engage in "reflective self-understanding" [27]. That is, the construct assumes that these qualities reflect a general self-regulatory capacity to experience one's actions and behaviors as self-organized or self-initiated [45].

Across both daily diary studies, we found convergent evidence that dispositional autonomy related to more self-determined motivation for solitude. The associations between

**Table 4. Random-intercept models that include introversion, dispositional autonomy, and all other big-five traits, predicting daily preference for solitude (Study 2).**

| Predictors | Preference for solitude | | | | | | Preference for solitude (controlling for gender & ethnicity) | | | | | |
|---|---|---|---|---|---|---|---|---|---|---|---|---|
| | B | CI(B) | ß | CI(ß) | t | p | B | CI(B) | ß | CI(ß) | t | p |
| (Intercept) | 4.46 | [2.96, 5.96] | .00 | [-.08, .08] | 5.82 | < **.001** | 4.75 | [3.19, 6.31] | -.05 | [-.24, .15] | 5.97 | < **.001** |
| Agreeableness | -.42 | [-.62, -.21] | -.18 | [-.27, -.09] | -3.96 | < **.001** | -.42 | [-.63, -.22] | -.19 | [-.28, -.10] | -4.03 | < **.001** |
| Neuroticism | -.11 | [-.28, .06] | -.06 | [-.14, .03] | -1.28 | .201 | -.15 | [-.33, .03] | -.08 | [-.17, .02] | -1.64 | .100 |
| Openness to experience | .05 | [-.17, .28] | .02 | [-.07, .11] | 0.46 | .647 | .17 | [-.04, .38] | .07 | [-.02, .16] | 1.56 | .119 |
| Conscientiousness | .17 | [-.03, .38] | .08 | [-.01, .17] | 1.67 | .095 | .12 | [-.03, .26] | .07 | [-.02, .15] | 1.57 | .116 |
| Introversion | .13 | [-.02, .28] | .08 | [-.01, .16] | 1.74 | .083 | .01 | [-.28, .30] | .00 | [-.09, .10] | 0.08 | .937 |
| Dispositional autonomy | .04 | [-.25, .33] | .01 | [-.08, .11] | 0.24 | .807 | .04 | [-.19, .26] | .01 | [-.08, .10] | 0.31 | .758 |
| Gender [woman vs. others] | | | | | | | .24 | [-.01, .50] | .18 | [-.01, .36] | 1.88 | .061 |
| Ethnicity [Asian vs. others] | | | | | | | -.05 | [-.35, .24] | -.04 | [-.25, .17] | -.37 | .713 |
| Ethnicity [White vs. others] | | | | | | | -.18 | [-.48, .11] | -.13 | [-.34, .08] | -1.21 | .228 |
| **Reliability estimates and estimates of between-person reliability and reliability of change** | | | | | | | | | | | | |

| | Variance | Percent |
|---|---|---|
| $\sigma^2_{Person}$ | 0.43 | .13 |
| $\sigma^2_{Day}$ | 0.01 | .00 |
| $\sigma^2_{Item}$ | 0.31 | .09 |
| $\sigma^2_{Person \times Day}$ | 0.99 | .29 |
| $\sigma^2_{Person \times Item}$ | 0.33 | .10 |
| $\sigma^2_{Day \times Item}$ | 0.01 | .00 |
| $\sigma^2_{Error}$ | 1.30 | .38 |

dispositional autonomy and self-determined motivation for solitude were consistently positive across the two studies. There were four models conducted across studies. In Study 1, we tested two models with the first one including only introversion and dispositional autonomy and the second one controlling for gender and ethnicity. In Study 2 we tested two models with the first one including introversion, dispositional autonomy, and other four Big-Five traits, and the second one controlling for gender and ethnicity. Out of four models, three yielded significant and positive associations between dispositional autonomy and self-determined motivation for solitude, with standardized coefficients ranging from .10 to .18. The effect sizes for the observed associations across both studies were small. In one model in Study 2 where we did not find significant association between these two variables, none of the other traits yielded significant associations, aside from a gender difference that emerged. We further explored whether gender interacted with personality traits to predict the two main outcome variables but did not find evidence for the moderationg effect. Since we did not conduct power analyses to determine our sample sizes for both studies, we conducted power calculation on simulated data. This is more appropriate than performing post-hoc power analyses, which have been criticized for not providing true observed power when being conducted on data that has been collected and analysed [46]. In a series of power simulations (*n* = 100 simulations) looking at achieved power across sample sizes between 140 and 270 and assuming 5 to 7 data points per participant, a sample of 270 (Study 2's sample) should allow 80% power to detect an effect size as small as .08. We included the R code of our power simulation in our OSF folder, along with the results of our simulation.

Overall, results point to dispositional autonomy as being a modest predictor of self-determined motivation for solitude. Individuals who rated themselves as higher in the tendency to be congruent in terms of how their behavior aligns with their values and interests, as resistant

to pressure from others, and as typically interested in learning more about their personal experiences and emotions, tended to approach alone time with a sense they were choicefully selecting it, and they saw this time as valuable and worthwhile. These findings shed new light on how individuals experience aloneness, further suggesting that the capacity to embrace and value time spent alone might be a function of how individuals self-regulate experiences and behaviors, more generally, rather than based more simply on preferences to be alone so as to avoid social interactions. Furthermore, they suggest that this self-regulation cannot be reduced to easily intuited personality traits such as introversion. In sum, the associations between dispositional autonomy and seeing time alone as enjoyable and valuable demonstrated the first evidence to show that solitary enjoyment related to the ability to regulate oneself in positive and self-congruent way.

These results applied an SDT perspective to better understand the relation between an autonomous personality and motivation for alone time. SDT's conceptualization of dispositional autonomy is different from Clark and Beck's [47] assertion about an autonomous person, one who feels more competent when achieving things alone and without the help of others. According to Clark and Beck, an autonomous personality tends to have a stronger preference for solitude and possesses a tendency to seek independence at any cost. This desire for independence and individual sense of accomplishment has been showed to breed dysphoria and loneliness [48]. In contrast to this view, dispositional autonomy from an SDT perspective is defined, as we describe above, in terms of the tendency and capacity to regulate one's experiences in accord with closely held interests and values, instead of because one feels pressured and forced to do things that are not fully endorsed. The SDT perspective of dispositional autonomy yields a different set of predictions concerning the quality of motivation for spending time alone. As such, our findings serve to clarify the distinction between these two above-described perspectives, and showed that dispositional autonomy rooted in the SDT framework relates to healthy motivation for spending time alone, which has been showed to correlate with relational and personal well-being [9]. In the same way that SDT's conceptualization of autonomous self-regulation does not concern with individual preferences [14], SDT's dispositional autonomy does not relate to preference for solitude.

Another important set of findings from these two studies was the absence of evidence supporting the common belief that those who are introverted like to spend time alone. Building on the literature suggesting that extroverts enjoy social attention and thus find more time to be around other people [49], it has only been inferred rather than empirically established that those high in introversion–who spend less time in social interactions–would enjoy being alone more [39]. This inference is rooted in the conception that the lack of social engagement is equivalent to enjoyment of the opposite state—aloneness. Our research findings contradicted this idea, and were different from the findings in a recent study by Thomas et al. [50]. In their study, the authors distinguished between high-functioning (happy) and low-functioning (not-happy) introverts, while our studies did not. Whereas we did not find an association between introversion and preference for solitude, they found that both types of introverts prefer solitude more than extraverts. Low-functioning introverts were motivated to seek out solitude for not self-determined reasons due to not feeling like they belong or fit into their peer groups. As was the case in our current studies, Thomas and Azmitia [1] also did not find evidence linking introversion and self-determined motivation for solitude.

Although the assumption that introverts like to spend time alone was not supported, there is more to explore. Future research might consider when and how extraverts like to have time alone. More importantly, our present findings highlighted that one cannot rely on psychological correlates of social engagement to infer the psychological correlates of solitary enjoyment. Solitude research may be more richly advanced if researchers study time alone as a unique

experience with its own particular correlates and dynamics, rather than as a derivative from the lack of enjoyment or opportunities for interpersonal interactions. In fact, just as a person can feel lonely both when spending time with others and when spending time alone [12], it is possible that personality characteristics that lead people to enjoy their time with others also predict their affinity for solitude, as was the case here for an autonomous disposition. To this end, our studies suggested that a healthy motivation for solitude might be related to an individual's tendency to regulate their daily experiences in autonomous and choiceful ways. We recognized that, while this point has been made theoretically [25], it is important to test it empirically and that is what we attempted here.

## Limitations and future directions

These findings should be understood in light of some methodological limitations. A first limitation is that we relied on an operationalization of introversion that has received several criticisms. Particularly, extraversion has historically been represented with more positively valanced items such as assertiveness, positive emotions, and excitement seeking [42]. This way of operationalizing the extraversion-introversion continuum also assumes that introversion lacks those qualities. To overcome this limitation, instead of using the NEO-PI that includes predominantly positive items to represent extraversion, we used the Big-Five Inventory, which combines both behavioral and affective contents to distinguish extraverts from introverts [42]. Nonetheless, it would be worthwhile for future research to test our hypotheses with different inventories of big-five traits.

A second limitation is that the daily diary method did not involve real-time assessments of solitary experiences. Instead, we measured participants' retrospective evaluations of individuals' preference and motivation for solitude at the end of each day. As such, it was not possible for us to investigate whether any situational or momentary factors that happen prior to solitude that could have contributed to participants' preference and motivation for this time. For example, our data suggested that some individuals' preference for solitude might vary depending on the day while others might prefer solitude more consistently across days. This means that, for some people, a moderator might explain why someone would prefer to spend time alone on a particular day, or that some kinds of events may enhance this preference. Research on solitude seeking after being ostracized [20, 51] could provide some insights here. It may be that individuals with certain personality traits might be more likely to prefer solitude after feeling ostracized or excluded by others.

A third limitation is that both studies relied on college or university samples. In this research, we recruited participants within the typical age range of those who attend universities in the United States, with a mean age of 20 years, lending caution to any generalizations to the larger population beyond young adults who attend higher education in the US. Whether education levels and different life circumstances might contribute to motivation and preference for solitude is an empirical question that warrants future research. Additionally, the dynamics of time spent alone change undoubtedly change as people get older. Older adults might find time alone more tolerable and positive [33, 34], and because of that, their personality characteristics might be more dissociated from their motivation for solitude. Attitudes toward time spent alone in older adults may also be more stable and positive over time because they have had more opportunities to structure their daily solitary experiences better in ways that work for their lifestyles and routines [33, 34].

Finally, cultural factors may also shape responding to alone experiences, for example in previous research individuals from Eastern cultures may, for example, perceive spending time alone to play different functions in their life than those from Western cultures [13]. Another

study also showed that self-determined motivation for solitude might be more salient for certain cultural groups than others [3]. Future research may benefit from examining these characteristics in more culturally diverse samples.

## Conclusion

Overall, the present findings across two diary studies showed that the motivation behind the time we spend alone can be affected by personality dispositions, though not necessarily the dispositions many have intuitively thought to be important. We specifically explored the role of *introversion*, thought by many to be associated with preferences for solitude, as well as individual differences in *autonomous regulation*–the propensities to behaving in self-congruent ways, take an interest in one's emotions, and feel free from pressure. We found consistent evidence that the latter personality characteristic links to the extent to which a person sees solitude as a positive and valuable experience–one that should be pursuit for its own right. Our findings suggested that while it does not incline individuals to prefer solitude over social time, a disposition toward autonomous regulation helps individuals to endorse the value of spending time with themselves in their everyday life.

## Author Contributions

**Conceptualization:** Thuy-vy T. Nguyen.

**Data curation:** Thuy-vy T. Nguyen.

**Formal analysis:** Thuy-vy T. Nguyen.

**Methodology:** Thuy-vy T. Nguyen.

**Supervision:** Netta Weinstein.

**Writing – original draft:** Thuy-vy T. Nguyen, Netta Weinstein.

**Writing – review & editing:** Netta Weinstein, Richard M. Ryan.

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
