## [Decision Letter · Decision Letter 0]

18 Jun 2021

PONE-D-21-07216

Who Enjoys Solitude? Autonomous Functioning (But Not Introversion) Predicts Self-Determined Motivation (But Not Preference) for Solitude

PLOS ONE

Dear Dr. Nguyen,

Thank you for submitting your manuscript to PLOS ONE. After careful consideration, we feel that the paper has merit but does not fully meet PLOS ONE’s publication criteria as it currently stands. Therefore, we invite you to submit a revised version of the manuscript that addresses the points raised during the review process.

We received reviews from 2 experts in the substantive topic and the methodological and data analytic approaches taken. Both reviewers found strengths with the manuscript but also concerns about a number of substantive points related to the study rationale, methodology, and analyses. I encourage you to give careful consideration to the reviewers’ thoughtful comments on your manuscript. My concerns are consistent with those of the reviewers, and I highlight several additional ones as well:

As we know, a disposition is a tendency to think, feel, and/or act in particular ways across a variety of situations. Thus it seems expectable that dispositional autonomy would predict daily self-determined motivation for spending time alone – that is, dispositional autonomy predicts autonomous behavior. Does this research contribute insights on autonomy and solitude beyond this expectable relation?
Concerning the samples: 1) How were sample sizes determined? Power analyses are needed here, or some other justification for the sample sizes. (2) Why were undergraduate students chosen as the population of interest (beyond convenience)?
In your tables, please specify whether you are presenting unstandardized (b) or standardized (beta) estimates. The tables present the symbol for the latter, but unless the data were standardized before analysis, I’m guessing the tables present b values.
On p. 14, there is an inconsistency between text and table in one of the lower bound CIs (-.006 versus .006). Please correct this. Also note that if the former bound is accurate, the relation you present is not reliable.
In Study 2, please clearly indicate whether and how the measures and procedure are the same as/different from those of Study 1.
In both studies, please control for sex and race in preliminary models or main analyses.

We look forward to receiving your revised manuscript.

Kind regards,

Kirk Warren Brown

Academic Editor

PLOS ONE

Journal Requirements:

2. Please change "female” or "male" to "woman” or "man" as appropriate, when used as a noun (see for instance https://apastyle.apa.org/style-grammar-guidelines/bias-free-language/gender).

*Please note that according to our submission guidelines (http://journals.plos.org/plosone/s/submission-guidelines), outmoded terms and potentially stigmatizing labels should be changed to more current, acceptable terminology. To this effect, please change 'Caucasian' to 'White' or 'of European ancestry.

3. In your Data Availability statement, you have not specified where the minimal data set underlying the results described in your manuscript can be found. PLOS defines a study's minimal data set as the underlying data used to reach the conclusions drawn in the manuscript and any additional data required to replicate the reported study findings in their entirety. All PLOS journals require that the minimal data set be made fully available. For more information about our data policy, please see http://journals.plos.org/plosone/s/data-availability. "Upon re-submitting your revised manuscript, please upload your study’s minimal underlying data set as either Supporting Information files or to a stable, public repository and include the relevant URLs, DOIs, or accession numbers within your revised cover letter. For a list of acceptable repositories, please see http://journals.plos.org/plosone/s/data-availability#loc-recommended-repositories. Any potentially identifying patient information must be fully anonymized. Important: If there are ethical or legal restrictions to sharing your data publicly, please explain these restrictions in detail. Please see our guidelines for more information on what we consider unacceptable restrictions to publicly sharing data: http://journals.plos.org/plosone/s/data-availability#loc-unacceptable-data-access-restrictions. Note that it is not acceptable for the authors to be the sole named individuals responsible for ensuring data access. We will update your Data Availability statement to reflect the information you provide in your cover letter.

Reviewers' comments:

Reviewer's Responses to Questions

**Comments to the Author**

1. Is the manuscript technically sound, and do the data support the conclusions?

Reviewer #1: Yes

Reviewer #2: Partly

2. Has the statistical analysis been performed appropriately and rigorously? 

Reviewer #1: Yes

Reviewer #2: No

3. Have the authors made all data underlying the findings in their manuscript fully available?

Reviewer #1: Yes

Reviewer #2: Yes

4. Is the manuscript presented in an intelligible fashion and written in standard English?

Reviewer #1: Yes

Reviewer #2: Yes

5. Review Comments to the Author

Reviewer #1: Review for PONE-D-21-07216 Who Enjoys Solitude? Autonomous Functioning (But Not Introversion) Predicts Self- Determined Motivation (But Not Preference) for Solitude

May 18, 2021

The authors have contributed a valuable addition to the empirical literature on solitude, which is sparse. In particular, they tackle the issue of differentiating the construct of solitude from the individual difference of introversion, which as they note has often been conflated, with mixed results from previous studies. In addition, their study asserts the importance of dispositional autonomy as a factor in solitude enjoyment, a factor that has largely been overlooked in the solitude literature. I have organized my comments according to the sections of the manuscript.

Introduction:

1. A correction on Thomas & Azmitia’s (2019) findings on p. 5. The authors state that Burger’s Preference for Solitude (PSS) measure showed negative correlation with self-determined motivation for solitude, but in actuality their results showed a significant positive correlation for both emerging adults and adolescents. The PSS also correlated positively with not-self-determined motivation, indicating that the PSS does not differentiate between intrinsic/autonomous/self-determined vs. extrinsic/non-autonomous/not-self-determined motivations.

2. Please cite the study in this sentence (p. 5): “In contrast, in another study that examined preference for solitude by Burger’s operationalization and self-determined motivation for solitude from SDT perspective in older adults showed positive correlations between these two concepts.”

3. It should be noted that Burger also found a significant positive correlation between introversion and preference for solitude in his original study (1995).

4. Can the authors state their hypothesis more clearly in the following sentence (p. 7): “We predicted that extraversion-introversion dimension of big-five personality would positively relate to preference for solitude but yield non-significant association with self-determined motivation for solitude.”

The phrase “extraversion-introversion dimension” renders the hypothesis unclear. Based on their introduction, I understand them to mean that introversion will positively correlate with preference for solitude; however, the Big 5 measures extraversion (high or low), thus a positive correlation would indicate that extraverts prefer solitude. It could be restated that extraversion scores will negatively correlate with preference for solitude (and motivation for solitude). Or, clearly state that you are reverse scoring the extraversion measure, so that all extravert items are reverse coded to yield an introvert score. You mention this briefly in the Methods section, but it is so non-traditional to frame this factor as Introversion that I think it would be worth explaining more clearly to the reader. On a related note, it is troubling that introversion on the Big 5 is so negatively valenced (defined with the items of shy, reserved, inhibited, as well as low scores on all of the “positive” items). I don’t know if this affects the authors’ interpretation of their findings, but it could be worth noting (there is a long tradition of such critiques of the Big 5 for this, and other reasons, e.g. Block 1995; 2010). This is not the authors’ problem, but it could be worth mentioning as a possible explanation for why laypeople and researchers alike regularly “intuit” that introversion must be related to solitude. Perhaps it is because the Big 5 is not accurately measuring introversion – only the negative side of low extraversion.

5. To support the authors’ framing of mixed results when it comes to introversion and solitude, the authors may want to include a more recent study by Thomas and colleagues (2020), in which cluster analysis differentiated between two clusters of introverts (high-functioning and low-functioning re: measures of adjustment), and these clusters did show different associations to preference for solitude, loneliness, and motivation for solitude.

6. The rationale for their second hypothesis (dispositional autonomy) is clear and well-reasoned.

Study 1

7. I appreciate that the authors utilized a daily diary methodology for this study, as I agree these types of methods improve ecological validity in general, and in particular for assessing how people experience their time alone.

8. A minor clarification - In Weinstein et al’s original study, the measure is called the Index of Autonomous Functioning (IAF) whereas the authors of this manuscript use the term Autonomous Functioning Index (AFI). I am not sure if the name of the scale has been altered since its original publication, or whether this is an error. Please revert to IAF if necessary.

9. A couple of comments on their preference for solitude measure. I found it odd that the authors did not use Burger’s Preference for Solitude Scale in their study after discussing it in their introduction. It can be difficult to compare results between this study and prior studies if the measures are not the same. It would be helpful if the authors could include an explanation for why his original 12-item scale was not used in their initial survey, along with the other personality measures. Second, it would be helpful if the authors could briefly explain how they selected the three questions they did use to assess preference for solitude during the daily diary study. For example, were their three questions inspired by, or derived from, Burger’s measure in any way? Were they theoretically generated based on other sources?

10. Please include the rating scale used for preference for solitude. Was it 1-5? This would help readers interpret the descriptive statistics on p. 13.

11. As with point 9 above, it would be helpful to briefly explain how they selected the four questions used to assess self-determined motivation for solitude during the daily diary study.

12. In the daily diary prompts, were the self-determined motivation questions only presented to participants if they had spent time alone that day? The way these questions are worded, it sounds as if they are referencing actual time the participant had spent alone.

13. Did the study collect data on the frequency or duration of time spent alone each day? If so, that would be valuable to add into the analysis.

Study 2

14. The inclusion of all 5 factors of the Big 5 Inventory is presented, convincingly, as a worthwhile addition in order to evaluate how influential dispositional autonomy is for predicting solitude behaviors.

15. It could be worth noting that agreeableness and conscientiousness were both marginally significant (<.10). They are very close to the .05 cutoff, and very close to the introversion results. In particular the results show a trend with low agreeableness being associated with preference and motivation for solitude. Theoretically this might make sense, given that highly agreeable people may be less able to resist social pressure (one of the aspects of the IAF), so people lower in agreeableness could potentially find it easier to withdraw from the social scene (potentially disrupting social norms or social harmony) and take their alone time.

Discussion

16. Change personality types to traits. (p. 20).

17. The stronger case for self-determined motivation for solitude and positive development was found with emerging adults rather than adolescents in the Thomas & Azmitia study they reference (p. 20).

18. I found the section (p. 21-22) following this sentence particularly strong and theoretically sound: In sum, the associations between dispositional autonomy and seeing time alone as enjoyable and valuable demonstrated the first evidence to show that solitary enjoyment related to the ability to regulate oneself in positive and self-congruent way.

19. Just a side note, re: a statement on p, 23 “…it is possible that the same personality characteristics that lead people to enjoy their time with others also predict their affinity for solitude.” Maslow made the essentially same point in his studies of self-actualization, when he noted that self-actualized people have both a love of privacy and a love of other people.

20. On p. 23, “To date, we are aware of no empirical data that has directly assessed the link between any personality characteristics…” but didn’t both Burger (1995) and Thomas and colleagues (2019; 2020) include introversion-extroversion data in relation to preference for solitude and motivation for solitude? See points 3 and 5, above. But, to the authors’ point, this study does appear to be the first that shows the links between the personality characteristic of autonomous functioning and solitude.

Proofreading:

The authors will want to proofread for typos, as there were many in the manuscript. A few examples, although I didn’t document them all: “I do thinks in order to avoid feeling ashamed” (p. 11); “Sixteen participants who filled out the initial survey did not the diary portion of the study…” (p. 16); “ First, all two studies relied on” (p. 23); etc.

Figures 1 and 2 have the same title. Can the authors add “in Study 1” to Figure 1 and “in Study 2” in Figure 2 (or something to that effect) so that the figures are differentiated?

Reviewer #2: This study differentiated between daily preference for solitude and daily self-determined motivation for solitude by linking the latter, and not the former, with trait-level autonomy. This study has the potential to contribute to the literature by demonstrating a link between self-determined motivation for solitude and the well-established personality factor of autonomy based on SDT. However, certain issues came up in my reading, mainly having to do with the data analysis, that I think would need to be addressed to prepare this paper for publication.

I commend the authors for posting their Rmarkdown code and data files on OSF. These were also very helpful to me in checking my own understanding of what the authors had done.

Introduction

1. Could the Introduction section explain further why introversion and autonomy were selected as measures for establishing discriminant validity of preference for solitude & self-determined motivation for solitude? (As opposed to other aspects of personality such as dispositional shyness, attachment styles, low sensation-seeking, etc.)? The Discussion section clearly explains the rationale for personality variable selection, but it is less clear from the Introduction section.

2. Since the Thomas & Azmitia motivation for solitude scale also includes a measure of not-self-determined motivation for solitude, I am curious why this scale was not included in the study (as another contrast to preference for solitude)?

3. I am a bit confused by the rationale for hypothesizing that introversion would show no association with self-determined motivation for solitude. The Leary et al. (2003) study cited in this section suggests the opposite, that that individuals low in characteristics related to extraversion/sociability/people orientation derive more enjoyment from engaging in activities alone than individuals high in these extraverted traits. Since other cited research is inconclusive on this, it seems that, on balance, the suggestion is that individuals higher in introversion might derive more enjoyment from solitary activities. What was the basis for the hypothesized null effect?

Method (Studies 1 and 2)

4. How was alpha (reliability) computed for daily-level preference for solitude and self-determined motivation for solitude? It appears from the study R code posted on OSF that Cronbach’s alpha is computed for level-1 and level-2 variables in the same manner. However, it is problematic to use Cronbach’s alpha directly on repeated measures data because it does not account for person-level clustering. The psych::multilevel.reliability function in R can be used to calculate reliability for a 2-level model (Nezlek, 2017).

Nezlek, J. B. (2017). A practical guide to understanding reliability in studies of within-person variability. Journal of Research in Personality, 69, 149-155.

5. Please specify the scale for the daily preference for solitude measure (Study 1 Method section). The description of the self-determined motivation for solitude measure is also not entirely clear. Did participants respond to this measure for each solitary event they reported, or was it administered once for the entire day? It is stated that “Scores on each item were averaged for both solitary episodes” so it sounds like two solitary episodes are used? (This is not explained)

6. The paper refers to random slopes for introversion, and random slopes for autonomy. However, in a model with 2 levels (level 1 = daily diaries, level 2 = people), level-2 predictors don’t have random slopes across people; rather, there is a fixed-effect slope that applies to the whole sample.

For example, for the autonomy-motivation for solitude slope, because each person only contributes a single autonomy score, it is not possible to compute a slope for each person. There is just an autonomy-motivation for solitude slope which applies to the entire sample, and which reflects the association between trait autonomy and each person’s mean motivation for solitude score across all study days. (If autonomy had been measured repeatedly at the daily level, it could have a random slope with motivation for solitude that varied across participants).

This leaves me confused about what is meant to be captured in Figures 1 and 2. If each person only provided one score for dispositional autonomy, how can a slope between this and any other variable be computed for a single person? It is possible that I am entirely misinterpreting what the authors intended to model.

In the R code posted on OSF, the level 2 variables have been added as random effects to this models; these models appear to have been mis-specified (or I have no idea what these random effects are modeling). Again, it is possible that I misunderstood what the authors intended to do.

7. More broadly, since the study hypotheses are not about any within-person associations, there seems to really be no need to do mixed-effects or multilevel modeling. Since all model predictors are at level-2 (person-level), their regression coefficients just reflect associations between the person-level predictor (e.g. trait autonomy) and each person’s mean of the daily-level outcome variable (e.g. each person’s mean motivation for solitude score across study days).

Results and Discussion

8. The results sections state for example that “Random-intercept models without fixed effects revealed an ICC of…” (p. 13). Do the authors mean “random-intercept models with no predictors”? When computing ICC, there should still be a fixed effect for the intercept. (The lmer function in R will also add a fixed intercept by default even if it’s not included in the model syntax).

9. For Study 2, did the authors have any specific hypotheses regarding associations between other personality factors (neuroticism, agreeableness, etc.) and daily measures of solitude preferences/motivation? The Discussion section should also say more about these associations and about the significant association between agreeableness and preference for solitude. Also, why was openness to experience excluded from the models reported in Tables 3 and 4?

10. The Results and Discussion sections refer to correlations between dispositional autonomy and self-determined motivation for solitude. Since mixed-effects regression models with control variables were used in analyses, it is not precise to refer to these relationships as correlations (this usually implies bivariate correlations at a single level). Terms such as “associations” or “relationships” could be used instead.

11. In the limitations section, it is stated, “Older adults might find time alone more tolerable (22), and therefore, their personality might affect motivation for solitude less strongly. Nonetheless, given that motivation and preference for solitude in young adults is more likely to vary daily whereas attitudes toward time spent alone in older adults remained more stable and positive over time…” Could the authors explain this further? Why would higher overall levels of preference for solitude in old age necessarily lead to reduced within-person variability in preference for solitude?

12. The study limitations section is limited in that in only mentions limitations related to the characteristics of the participant sample. Could the authors also address potential limitations of the study design, measures, etc.?

Minor comments

13. The Methods and Results section headings should specify whether they’re referring to Study 1 or Study 2 (to avoid ambiguity)

14. On p. 5: “In contrast, in another study that examined preference for solitude by Burger’s operationalization and self-determined motivation for solitude from SDT perspective in older adults showed positive correlations between these two concepts.” Which study is this sentence referring to? And in which age group? The implication in this paragraph is that mixed findings may be due to different motivations across age groups, but it is not clear exactly where this conclusion comes from.

15. In the study 1 Participants section (p. 9), it is stated “One hundred and eighty three undergraduate students (153 females; 9% Hispanic), and then it is stated, “The sample consisted of 53% Whites and Caucasians, 35% Asians or Pacific Islanders, 5% Black or African Americans, …” This double description of the sample ethnic make-up is a bit confusing as they do not match.

6. PLOS authors have the option to publish the peer review history of their article (what does this mean?). If published, this will include your full peer review and any attached files.

Reviewer #1: No

Reviewer #2: No

---

## [Author Response · Author response to Decision Letter 0]

15 Nov 2021

Dear Dr. Brown, 

Thank you for your feedback on our earlier draft of the manuscript: “Who Enjoys Solitude? Autonomous Functioning (But Not Introversion) Predicts Self-Determined Motivation (But Not Preference) for Solitude” (PONE-D-21-07216) and for soliciting feedback from the two expert reviewers. Your and reviewer comments were very helpful, and we have thought carefully about each and how to best undertake any improvements to the paper. We detail each below in bolded font, and our response to it.

Editor Comments

We received reviews from 2 experts in the substantive topic and the methodological and data analytic approaches taken. Both reviewers found strengths with the manuscript but also concerns about a number of substantive points related to the study rationale, methodology, and analyses. I encourage you to give careful consideration to the reviewers’ thoughtful comments on your manuscript. My concerns are consistent with those of the reviewers, and I highlight several additional ones as well:

1. As we know, a disposition is a tendency to think, feel, and/or act in particular ways across a variety of situations. Thus it seems expectable that dispositional autonomy would predict daily self-determined motivation for spending time alone – that is, dispositional autonomy predicts autonomous behavior. Does this research contribute insights on autonomy and solitude beyond this expectable relation?

Response. Thank you for this comment. We thought long and hard about the implications of this issue, as it speaks directly to how clearly we had communicated the theoretical contribution of the paper in the previous version of the manuscript. We concluded that two elements of the work must be brought out, and have made some revision in the Introduction to emphasize the following contributions: 

First, our paper was the first to explicitly state the conceptual distinction between the two concepts, preference for solitude and self-determined motivation for solitude. The two literatures have existed in parallel, but though they are understood to be conceptually distinct, they have not been tested in parallel for the purpose of clarifying how or why they are different. We have tried to do that here by examining their distinct relations with personality predictors. To help make this point, we have added text throughout the Introduction and Discussion. For example, we now start with an organizing paragraph new to this version: 

“Evidence is mounting that self-determined motivation for solitude is empirically distinct from merely preferring solitude over social time (1, 9, 10), but there is little knowledge about what drives people to self-determined motivation for solitude, even when they do not necessarily prefer to be alone. In this paper, we explore the role that personality plays in both motivation and preference for solitude, focusing on two personality characteristics that should predict the two constructs differentially: autonomous orientation, which is likely to drive self-determined motivation, and introversion, likely to drive preference” (p. 3)

Second, we have added the paragraph below to the Introduction to highlight that the conceptual work which aims to understand whether a disposition that differentiate between those who self-regulate and those who do not (i.e., autonomous orientation), can explain healthy motivation toward a self-connecting experience that is commonly portrayed as challenging (with solitude being a prime example). 

“Previous literature has demonstrated that those who tend behave in ways that are consistent with their beliefs and values tend to be motivated by intrinsic and self-determined reasons in whatever they do (36,37,39). However, it is important to highlight that solitude is commonly portrayed as a challenging experience for people. This is particularly relevant for young adults because they find solitude more difficult than older age groups (24,40) unless they have been allowed opportunities to develop a capacity to enjoy it. Given this, an investigation of dispositional autonomy that represents that capacity speaks to Winnicott’s theorizing (32) discussed above, which has not been considered in previous studies.”(p. x)

In this paper, we sought to highlight the theoretical underpinning of this investigation – the proposition of Donald Winnicott around the capacity to be alone and how this capacity can be nurtured through development. We believe this work investigates a concept that is similar to that capacity of which Winnicott speaks. We explained Donald Winnicott’s capacity to be alone in an earlier section of the Introduction on p. 8. 

2. Concerning the samples: 1) How were sample sizes determined? Power analyses are needed here, or some other justification for the sample sizes. (2) Why were undergraduate students chosen as the population of interest (beyond convenience)?

Response. We now state in the Method of Studies 1 (p. 10) and 2 (p. 17) the rationales behind our sample size. We believe it is justified to determine sample sizes based on realistic measures: expectations of how many participants we can recruit from the pool and lack of external funding to recruit outside of this pool. 

We also wrote now: “We selected to test study hypotheses with students because we could access and track this population; however, it is worth noting that this age group – representing emerging adulthood – have been the focus of past research which has found robust links between self-determined motivation and well-being in solitude (Nguyen et al., 2018; Thomas & Azmitia, 2018), giving added value to our current tests of antecedents of self-determined motivation.” (p. 10)

3. In your tables, please specify whether you are presenting unstandardized (b) or standardized (beta) estimates. The tables present the symbol for the latter, but unless the data were standardized before analysis, I’m guessing the tables present b values.

Response. We now present both unstandardized (b) and standardized (beta or ß) estimates within the tables. 

4. On p. 14, there is an inconsistency between text and table in one of the lower bound CIs (-.006 versus .006). Please correct this. Also note that if the former bound is accurate, the relation you present is not reliable.

Response. Thank you for identifying this error. You are correct – both should have been .006. However, following Reviewer 2’s comment, we have fixed the errors in our analyses – removing random slopes because they should not be included in the models. Therefore, we have produced new tables (Tables 1-4) which present the main analyses of the two studies. 

5. In Study 2, please clearly indicate whether and how the measures and procedure are the same as/different from those of Study 1

Response. On p. 17 of the manuscript, we now clarify that the measures and procedures are identical to those of Study 1, except that in study 2 we measured all five personality traits. 

6. In both studies, please control for sex and race in preliminary models or main analyses.

Response. We have now added gender and ethnicity to the secondary models (presented on the right side of each Table 1-4), with these two predictors included as covariates in all the confirmatory analyses. 

REVIEWER 1

The authors have contributed a valuable addition to the empirical literature on solitude, which is sparse. In particular, they tackle the issue of differentiating the construct of solitude from the individual difference of introversion, which as they note has often been conflated, with mixed results from previous studies. In addition, their study asserts the importance of dispositional autonomy as a factor in solitude enjoyment, a factor that has largely been overlooked in the solitude literature. I have organized my comments according to the sections of the manuscript.

1. A correction on Thomas & Azmitia’s (2019) findings on p. 5. The authors state that Burger’s Preference for Solitude (PSS) measure showed negative correlation with self-determined motivation for solitude, but in actuality their results showed a significant positive correlation for both emerging adults and adolescents. The PSS also correlated positively with not-self-determined motivation, indicating that the PSS does not differentiate between intrinsic/autonomous/self-determined vs. extrinsic/non-autonomous/not-self-determined motivations.

Response. Thank you for this catch. We have now made this change on p. 6 of the manuscript. It now reads: “Burger’s 12-item measure of preference for solitude (17) showed positive correlations with both measures that Thomas and Azmitia (1) used to assess self-determined and not-self-determined motivation for solitude in late adolescence. Another study that examined preference for solitude in adults older than 35 years of age showed positive correlation between Burger’s preference for solitude scale and measure of self-determined motivation for solitude, yet negative correlation between preference and measure of not self-determined motivation for solitude (23). These correlations suggest that, for those beyond adolescence and young adulthood, preference for solitude is associated more with self-determined rather than not-self-determined motivation for solitude, whereas preference for solitude in late adolescence could reflect both types of motivation. From this set of mixed results, it is possible that preference for solitude is motivated differently across age groups (24) and might have varied psychosocial implications for the individuals that prefer to be alone (25).”

2. Please cite the study in this sentence (p. 5): “In contrast, in another study that examined preference for solitude by Burger’s operationalization and self-determined motivation for solitude from SDT perspective in older adults showed positive correlations between these two concepts.”

Response. We now cited the sentence referring to the work by Weinstein and Nguyen (2021) (citation #10) which included the correlations between preference for solitude and both types of motivation (i.e., identified as equivalent to self-determined motivation for solitude, external as equivalent to not-self-determined motivation for solitude).

3. It should be noted that Burger also found a significant positive correlation between introversion and preference for solitude in his original study (1995).

Response. We have now included Burger’s finding concerning the significant positive correlation to the Introduction under the section discussing the association between introversion and preference for solitude on p. 6 (citation #17). 

4. Can the authors state their hypothesis more clearly in the following sentence (p. 7): “We predicted that extraversion-introversion dimension of big-five personality would positively relate to preference for solitude but yield non-significant association with self-determined motivation for solitude.”

The phrase “extraversion-introversion dimension” renders the hypothesis unclear. Based on their introduction, I understand them to mean that introversion will positively correlate with preference for solitude; however, the Big 5 measures extraversion (high or low), thus a positive correlation would indicate that extraverts prefer solitude. It could be restated that extraversion scores will negatively correlate with preference for solitude (and motivation for solitude). Or, clearly state that you are reverse scoring the extraversion measure, so that all extravert items are reverse coded to yield an introvert score. You mention this briefly in the Methods section, but it is so non-traditional to frame this factor as Introversion that I think it would be worth explaining more clearly to the reader. 

Response. We have now rewritten the hypothesis so that it is clearer: “we predicted that higher scores on introversion would positively relate to higher ratings of daily preference for solitude. In this research, we operationalized introversion as having the opposites qualities with those that are often associated with extraversion; that is, introverts are someone who tend to be more reserved, quieter, less talkative, energetic, or assertive. We will describe how we assess introversion in the method section.” (p. 10).

On a related note, it is troubling that introversion on the Big 5 is so negatively valenced (defined with the items of shy, reserved, inhibited, as well as low scores on all of the “positive” items). I don’t know if this affects the authors’ interpretation of their findings, but it could be worth noting (there is a long tradition of such critiques of the Big 5 for this, and other reasons, e.g. Block 1995; 2010). This is not the authors’ problem, but it could be worth mentioning as a possible explanation for why laypeople and researchers alike regularly “intuit” that introversion must be related to solitude. Perhaps it is because the Big 5 is not accurately measuring introversion – only the negative side of low extraversion.

Response. The reviewer is absolutely correct about this typical problem with operationalizations of introversion. We have now clarified our own operationalization and measure of introversion in two places, which includes both positive and negative characteristics:

1. One at the end of p. 10 in the Introduction where we state our hypothesis for introversion: 

“In this research, we operationalized introversion as having the opposite qualities with those that are often associated with extraversion; that is, introverts are someone who tend to be more reserved, quieter, less talkative, energetic, or assertive. We will describe how we assess introversion in the method section.”

2. The second place is in the Method section, on pp. 13-14, where we describe our measure: 

“To measure introversion, we used the eight items from John and Srivastava’s (41) Big Five Inventory (BFI) that were used to measure the big-five trait of extraversion. This measure includes descriptive statements of how extraverts often behave, such as being “talkative” and “full of energy”. We used this instead of the NEO-PI (Neuroticism, Extraversion, Openness Personality Inventory, 41) because the NEO-PI includes positive emotions and warmth as descriptors of extraversion, which could potentially portray introversion as lacking those positive qualities (see comparison between NEO-PI and BFI in Zillig et al (42)). Using the BFI Extraversion subscale, we showed participants a stem stating, “I see myself as someone who…”, and they proceeded to rate their agreement to a series of descriptive statements. We reverse-coded the extraversion-related items like “talkative” and “full of energy”, and averaged them with items included “reserved” and “quiet”, such that higher overall scores reflected introversion (α = .87). Participants rated their responses on a 5-point Likert scale, ranging from 1 = “strongly disagree” to 5 = “strongly agree”. ”

3. The third place is in the Limitations and Future Direction, on p. 26, where we addressed the limitation of existing measures of extraversion-introversion dimension:

“A first limitation is that we relied on an operationalization of introversion that has received several criticisms. Particularly, extraversion has historically been represented with more positively valanced items such as assertiveness, positive emotions, and excitement seeking (42); this assumes introversion lacks those qualities. To overcome this limitation, instead of using the NEO-PI that includes predominantly positive items to represent extraversion, we used the Big-Five Inventory, which combines both behavioral and affective contents to distinguish extraverts from introverts (42). Nonetheless, it would be worthwhile for future research to test our hypotheses with a wide range of different inventories of big-five traits.”

5. To support the authors’ framing of mixed results when it comes to introversion and solitude, the authors may want to include a more recent study by Thomas and colleagues (2020), in which cluster analysis differentiated between two clusters of introverts (high-functioning and low-functioning re: measures of adjustment), and these clusters did show different associations to preference for solitude, loneliness, and motivation for solitude.

Response. We agree that Thomas et al. (2020) would be a valuable addition to this paper. We have now cited the authors in the Discussion. We see that Thomas et al. (2020)’s findings are still different from our finding. Although we did not find the association between introversion and preference for solitude, Thomas et al. (2020) found that both low- and high-functioning introvert clusters had higher scores on preference for solitude than the extravert cluster. Therefore, we included this in the Discussion on p. 24 as follow:

“Our research contradicted such misconception and were different from the findings by a recent study by Thomas et al (50). In their study, they distinguished between high-functioning (happy introverts) and low-functioning introverts (not-happy introverts), while our studies did not. Whereas we did not find correlation between introversion and preference for solitude, they found that both types of introverts prefer solitude more than extraverts. Low-functioning introverts were motivated to seek out solitude for not self-determined reasons due to not feeling like they belong or fit into their peer groups. However, their findings also did not find evidence that both groups of introverts showed more self-determined motivation for solitude than extraverts.”

7. I appreciate that the authors utilized a daily diary methodology for this study, as I agree these types of methods improve ecological validity in general, and in particular for assessing how people experience their time alone.

Response. We are glad that the reviewer agrees with our choice of research design as we believe this methodology allows for more precise assessments of people’s daily experiences with solitude. 

8. A minor clarification - In Weinstein et al’s original study, the measure is called the Index of Autonomous Functioning (IAF) whereas the authors of this manuscript use the term Autonomous Functioning Index (AFI). I am not sure if the name of the scale has been altered since its original publication, or whether this is an error. Please revert to IAF if necessary.

Response. We have now changed the name of the scale to the original IAF.

9. A couple of comments on their preference for solitude measure. I found it odd that the authors did not use Burger’s Preference for Solitude Scale in their study after discussing it in their introduction. It can be difficult to compare results between this study and prior studies if the measures are not the same. It would be helpful if the authors could include an explanation for why his original 12-item scale was not used in their initial survey, along with the other personality measures. Second, it would be helpful if the authors could briefly explain how they selected the three questions they did use to assess preference for solitude during the daily diary study. For example, were their three questions inspired by, or derived from, Burger’s measure in any way? Were they theoretically generated based on other sources?

Response. We have now elaborated on the existing text to describe the source of items and our decision-making process in selecting this source within the Method section on p. 14. To summarize here, we were inspired by a short measure used by Wang et al. (2013) and felt that their items were more appropriate for our daily diary design and operationalization of preference for solitude; that is, we are interested in whether participants would rather spend time alone more than with other people on a specific day. We felt that Burger’s measure conflates preference and desire with enjoyment and benefits of solitude. This is the paragraph we added to p. 14:

“We decided to use this measure rather than Burger’s preference for solitude (17) because Wang et al.’s measure is shorter and thus more appropriate for daily diary design. Further, Burger’s measure includes some items that describe specific situations that might not apply to our participants’ daily experiences, such as “I like to vacation in places where there are few people around and a lot of serenity and quiet” or “If I were to take a several-hour plane trip, I would like to sit next to someone who was pleasant to talk with”. Some of Burger’s items also conflate preference and desire to be alone with enjoyment and benefits of being alone (i.e., “I enjoy being by myself”, Time spent alone is often productive for me”). Because we aimed to distinguish preference and desire from motivation to be alone for the enjoyment and benefits of solitude, we opted not to include those items in our measure.”

10. Please include the rating scale used for preference for solitude. Was it 1-5? This would help readers interpret the descriptive statistics on p. 13.

Response. We have now added this detail to the description of the scale. The items were rated on 7-point scale (from 1 = not at all true to 7 = very true) to keep it consistent with the measure of self-determined motivation for solitude. 

11. As with point 9 above, it would be helpful to briefly explain how they selected the four questions used to assess self-determined motivation for solitude during the daily diary study.

Response. We have now described the source of items for assessing self-determined motivation for solitude in the Method section on p. 15. Namely, we used the items developed by Nguyen et al. (2018) rather than Thomas and Azmitia (2019)’s measure because it offered a better fit for measuring daily solitude expeirence. We detail the reasons for this in the same paragraph: 

“For this variable, we used the scale from Nguyen et al. (2), which we felt was more appropriate for a diary study design because it measures state-level motivation for solitude and could better capture day-to-day fluctuation. In comparison, the measure developed by Thomas and Azmitia (1) is more appropriate for distinguishing individual differences in self-determined motivation for solitude.” (p. 14)

12. In the daily diary prompts, were the self-determined motivation questions only presented to participants if they had spent time alone that day? The way these questions are worded, it sounds as if they are referencing actual time the participant had spent alone.

Response. Questions assessing self-determined motivation asked participants about all the instances they were by themselves on that day, so they were not about any specific time participants had spent alone. The participants were presented with a prompt, which has now been added to the description of the scale on p. 14: “Different people spend time by themselves for different reasons. Please indicate the extent to which each of the following reasons applies to you regarding all the instances when you were by yourself today”

13. Did the study collect data on the frequency or duration of time spent alone each day? If so, that would be valuable to add into the analysis.

Response. The study did not involve data collection on frequency or duration of time spent alone. We were mainly interested in the two measures of preference for solitude and self-determined motivation for solitude. 

14. The inclusion of all 5 factors of the Big 5 Inventory is presented, convincingly, as a worthwhile addition in order to evaluate how influential dispositional autonomy is for predicting solitude behaviors.

Response. Thank you for this comment.

15. It could be worth noting that agreeableness and conscientiousness were both marginally significant (<.10). They are very close to the .05 cutoff, and very close to the introversion results. In particular the results show a trend with low agreeableness being associated with preference and motivation for solitude. Theoretically this might make sense, given that highly agreeable people may be less able to resist social pressure (one of the aspects of the IAF), so people lower in agreeableness could potentially find it easier to withdraw from the social scene (potentially disrupting social norms or social harmony) and take their alone time.

Response. We thank the Reviewer for this suggestion. However, we opted not to interpret marginally significant results, particularly for associations that we did not a-priori hypotheses for. We made this choice to avoid over-interpreting results that could be spurious. 

16. Change personality types to traits. (p. 20).

Response. We have made this change. 

17. The stronger case for self-determined motivation for solitude and positive development was found with emerging adults rather than adolescents in the Thomas & Azmitia study they reference (p. 20).

Response. Thank you for spotting this. We have made this change on p. 8, referencing Thomas & Azmitia (2019)

“Indeed, this view of healthy motivation for solitude has been reflected in recent empirical works, which showed that the pursuit of solitude for its benefits to creativity and relaxation was linked to experiences of personal growth in young adults (1).”

18. I found the section (p. 21-22) following this sentence particularly strong and theoretically sound: In sum, the associations between dispositional autonomy and seeing time alone as enjoyable and valuable demonstrated the first evidence to show that solitary enjoyment related to the ability to regulate oneself in positive and self-congruent way.

Response. Thank you for this. We are glad the Reviewer agrees with our interpretation. 

19. Just a side note, re: a statement on p, 23 “…it is possible that the same personality characteristics that lead people to enjoy their time with others also predict their affinity for solitude.” Maslow made the essentially same point in his studies of self-actualization, when he noted that self-actualized people have both a love of privacy and a love of other people.

Response. Thank you for this positive comment. We agree that dispositional autonomy is very conceptually close to the notion of self-actualized person by Maslow. In fact, Maslow’s theory around self-actualization was cited often in self-determination theory works and contributed to SDT formation. The argument of how the two conceptualizations of an autonomous person and a self-actuazlied person are connected has been demonstrated elsewhere (e.g., Sheldon & Kasser, 1995) and is outside the scope of this paper so we have opted not to refer to Maslow in our manuscript.

Sheldon, K. M., & Kasser, T. (1995). Coherence and congruence: Two aspects of personality integration. Journal of personality and social psychology, 68(3), 531.

20. On p. 23, “To date, we are aware of no empirical data that has directly assessed the link between any personality characteristics…” but didn’t both Burger (1995) and Thomas and colleagues (2019; 2020) include introversion-extroversion data in relation to preference for solitude and motivation for solitude? See points 3 and 5, above. But, to the authors’ point, this study does appear to be the first that shows the links between the personality characteristic of autonomous functioning and solitude.

Response. We have now removed this sentence. 

The authors will want to proofread for typos, as there were many in the manuscript. A few examples, although I didn’t document them all: “I do thinks in order to avoid feeling ashamed” (p. 11); “Sixteen participants who filled out the initial survey did not the diary portion of the study…” (p. 16); “ First, all two studies relied on” (p. 23); etc.

Figures 1 and 2 have the same title. Can the authors add “in Study 1” to Figure 1 and “in Study 2” in Figure 2 (or something to that effect) so that the figures are differentiated?

Response. Following Reviewer 2’s comments (described below), Figure 1 and Figure 2 are no longer needed for this manuscript.

REVIEWER 2

This study differentiated between daily preference for solitude and daily self-determined motivation for solitude by linking the latter, and not the former, with trait-level autonomy. This study has the potential to contribute to the literature by demonstrating a link between self-determined motivation for solitude and the well-established personality factor of autonomy based on SDT. However, certain issues came up in my reading, mainly having to do with the data analysis, that I think would need to be addressed to prepare this paper for publication.

I commend the authors for posting their Rmarkdown code and data files on OSF. These were also very helpful to me in checking my own understanding of what the authors had done.

1. Could the Introduction section explain further why introversion and autonomy were selected as measures for establishing discriminant validity of preference for solitude & self-determined motivation for solitude? (As opposed to other aspects of personality such as dispositional shyness, attachment styles, low sensation-seeking, etc.)? The Discussion section clearly explains the rationale for personality variable selection, but it is less clear from the Introduction section.

Response. We now have elaborated in the Introduction more clearly why we were interested in comparing the different effects of both introversion and dispositional autonomy (also in response to Editor Comment 1). Along with a new opening paragraph (p. 3), the rationale for testing introversion was explained in pp. 6-7. For one, we highlight the mixed findings in previous literature regarding the association between introversion and perceiving solitude as enjoyable. Based on these mixed results, we predicted the positive correlation between introversion and preference for solitude, but made no such prediction for self-determined motivation for solitude. The rationale for looking at dispositional autonomy was explained on pp. 7-9. To do so, we described theoretical views of Donald Winnicott regarding the capacity to be alone (p. 7), and we also highlight why autonomous disposition, here tested through interest-taking, self-congruence, and freedom from pressure may make it easier and more appealing to be alone with one’self’ (p. 8).

2. Since the Thomas & Azmitia motivation for solitude scale also includes a measure of not-self-determined motivation for solitude, I am curious why this scale was not included in the study (as another contrast to preference for solitude)?

Response. We had some hesitation around the measure of not self-determined motivation for solitude by Thomas and Azmitia because this measure include items that might touch on social anxiety (i.e., “I feel anxious when I’m with others”), and lack of belonging (i.e. “I don’t feel liked when I’m with others”, “I feel like I don’t belong when I’m with others”). That framing is not within the scope of this manuscript, which was instead interested more specifically in seeing value and personal reward in solitude. This is the reason we did not use that subscale and focused on distinguished between preference for solitude and self-determined motivation for solitude. 

3. I am a bit confused by the rationale for hypothesizing that introversion would show no association with self-determined motivation for solitude. The Leary et al. (2003) study cited in this section suggests the opposite, that that individuals low in characteristics related to extraversion/sociability/people orientation derive more enjoyment from engaging in activities alone than individuals high in these extraverted traits. Since other cited research is inconclusive on this, it seems that, on balance, the suggestion is that individuals higher in introversion might derive more enjoyment from solitary activities. What was the basis for the hypothesized null effect?

Response. The rationales for not predicting the association between introversion and self-determined motivation for solitude were explained on p. 10, after highlighting the theoretical argument for why introversion may relate to preference, but not self-determined motivation, for solitude.

To do this, we described the findings that showed positive correlation between introversion and preference for solitude and general positive attitudes toward solitude in cross-sectional design. However, these findings were interpreted to mean that introverts enjoy solitude more than extraverts. Then, we described findings that tracked day-to-day experiences with solitude and showed that extraverts actually rated both social and solitary experiences positively, and introverts also felt less positive than extraverts when not interacting with other people. Therefore, introverts might spend more time alone and might generally report preferring being alone more, that does not mean they enjoy solitude more or seek it out because of its enjoyment. On p. 10 we write: 

“However, while Burger (17) and Leary et al. (21) used one-time measures of people’s evaluations of their own personality and attitudes toward solitude, a study by Srivastava et al. (29), which used day-construction design and collected data from participants’ day-to-day experiences, showed a different pattern. When asked to report levels of positive affect when interacting with other people and when not interacting, those high in extraversion reported feeling more positive in both types of experiences. Interestingly, those low in extraversion also felt more positive in social interaction than when not interacting, and in fact felt less positive than extraverts when not interacting with others. This presents an interesting picture for introverts when it comes to their time spent alone; they generally show a greater preference for solitude than extraverts, (1) because they do not derive as much benefit out of social interactions as extraverts, but they might not necessarily enjoy time alone more (30). In other words, introverts’ preference for solitude might be driven more by the lack of appeal held by available social experiences, and less by their anticipation that spending time alone would be enjoyable.”

4. How was alpha (reliability) computed for daily-level preference for solitude and self-determined motivation for solitude? It appears from the study R code posted on OSF that Cronbach’s alpha is computed for level-1 and level-2 variables in the same manner. However, it is problematic to use Cronbach’s alpha directly on repeated measures data because it does not account for person-level clustering. The psych::multilevel.reliability function in R can be used to calculate reliability for a 2-level model (Nezlek, 2017).

Nezlek, J. B. (2017). A practical guide to understanding reliability in studies of within-person variability. Journal of Research in Personality, 69, 149-155.

Response. We greatly appreciated this suggestion by the Reviewer. We have now reported reliability across all items and assessment (days within person) for preference for solitude and self-determined motivation for solitude. Further, we also included Scale Reliability section for both studies to describe the variance components at different levels (i.e., item, day, person). You can also see percentages of variance at each level in all the tables (Table 1-2 for Study 1; Tables 3-4 for Study 2). 

5. Please specify the scale for the daily preference for solitude measure (Study 1 Method section). The description of the self-determined motivation for solitude measure is also not entirely clear. Did participants respond to this measure for each solitary event they reported, or was it administered once for the entire day? It is stated that “Scores on each item were averaged for both solitary episodes” so it sounds like two solitary episodes are used? (This is not explained)

Response. We have now added this detail to the description of the scale. Specifically, we describe that the items were rated on 7-point scale (from 1 = not at all true to 7 = very true) to keep it consistent with the measure of self-determined motivation for solitude. Participants responded to items in both measures once for the entire day; that means, the measures assess participants’ evaluations of their solitary experiences in general on each day. The statement “Scores on each item were averaged for both solitary episodes” was included by mistake. We apologized for this mistake and were grateful that the Reviewer had picked up this error; we have checked our codebook and removed this statement.

6. The paper refers to random slopes for introversion, and random slopes for autonomy. However, in a model with 2 levels (level 1 = daily diaries, level 2 = people), level-2 predictors don’t have random slopes across people; rather, there is a fixed-effect slope that applies to the whole sample.

For example, for the autonomy-motivation for solitude slope, because each person only contributes a single autonomy score, it is not possible to compute a slope for each person. There is just an autonomy-motivation for solitude slope which applies to the entire sample, and which reflects the association between trait autonomy and each person’s mean motivation for solitude score across all study days. (If autonomy had been measured repeatedly at the daily level, it could have a random slope with motivation for solitude that varied across participants).

This leaves me confused about what is meant to be captured in Figures 1 and 2. If each person only provided one score for dispositional autonomy, how can a slope between this and any other variable be computed for a single person? It is possible that I am entirely misinterpreting what the authors intended to model.

In the R code posted on OSF, the level 2 variables have been added as random effects to this models; these models appear to have been mis-specified (or I have no idea what these random effects are modeling). Again, it is possible that I misunderstood what the authors intended to do.

Response. Again, we are grateful for this comment because the Reviewer was right to point out that random slopes should not be included, since there is no way for each participant to have their own slope for personality differences predicting daily preference and motivation. This was entirely our misunderstanding of the data structure. We have consulted with a statistics expert that is more familiar with multilevel data and they have made the same suggestion: to only conduct random-intercept models for both outcomes. We have corrected this accordingly. You now can find our description of the models for Study 1 on p. 17 and Study 2 on p. 20. 

7. More broadly, since the study hypotheses are not about any within-person associations, there seems to really be no need to do mixed-effects or multilevel modeling. Since all model predictors are at level-2 (person-level), their regression coefficients just reflect associations between the person-level predictor (e.g. trait autonomy) and each person’s mean of the daily-level outcome variable (e.g. each person’s mean motivation for solitude score across study days).

Response. This is correct. We have corrected our description of the models and also reran the analyses. The new analyses do not change the conclusions of the study, and new analyses are now summarized in Tables 1 and 4 and described in the text of Study 1 (p. 16) and 2 (p. 18) Results sections. 

8. The results sections state for example that “Random-intercept models without fixed effects revealed an ICC of…” (p. 13). Do the authors mean “random-intercept models with no predictors”? When computing ICC, there should still be a fixed effect for the intercept. (The lmer function in R will also add a fixed intercept by default even if it’s not included in the model syntax).

Response. Thank you for this. We have now removed this statement. 

9. For Study 2, did the authors have any specific hypotheses regarding associations between other personality factors (neuroticism, agreeableness, etc.) and daily measures of solitude preferences/motivation? The Discussion section should also say more about these associations and about the significant association between agreeableness and preference for solitude. Also, why was openness to experience excluded from the models reported in Tables 3 and 4?

Response. We did not have any specific hypotheses regarding associations between other big-5 personality factors. We included all big-5 traits in Study 2 following a review round at the journal where we previously submitted this work as the previous reviewers felt that it would give stronger evidence for whether these two characteristics of interest predicted the outcomes when all big-5 traits were included. Reviewer 1 also agreed with this decision and thought the addition of all big-5 factors was appropriate. However, because we did not make any a-priori predictions for those factors, we preferred not to try and interpret them after the fact. 

Furthermore, the missing of openness in one of the models was a mistake on our part. Thank you for spotting this. We now have added results of openness to Tables 3 and 4. 

10. The Results and Discussion sections refer to correlations between dispositional autonomy and self-determined motivation for solitude. Since mixed-effects regression models with control variables were used in analyses, it is not precise to refer to these relationships as correlations (this usually implies bivariate correlations at a single level). Terms such as “associations” or “relationships” could be used instead.

Response. We have now used the term “association” instead of “correlation” to describe our results. 

11. In the limitations section, it is stated, “Older adults might find time alone more tolerable (22), and therefore, their personality might affect motivation for solitude less strongly. Nonetheless, given that motivation and preference for solitude in young adults is more likely to vary daily whereas attitudes toward time spent alone in older adults remained more stable and positive over time…” Could the authors explain this further? Why would higher overall levels of preference for solitude in old age necessarily lead to reduced within-person variability in preference for solitude?

Response. We now have added this explanation to clarify the age differences (p. x): 

“Nonetheless, given that motivation and preference for solitude in young adults are more likely to vary daily depending situations preceding solitude (i.e., after being ostracized; 18), the samples or our studies were appropriate for an investigation of how motivation and preference fluctuate on a day-to-day basis. On the other hand, attitudes toward time spent alone in older adults remained more stable and positive over time because they have had more opportunities to structure their daily solitary experiences better in ways that work for their lifestyles and routines (24).”

12. The study limitations section is limited in that in only mentions limitations related to the characteristics of the participant sample. Could the authors also address potential limitations of the study design, measures, etc.?

Response. We have now done this. Specifically, we discussed the limitation of not including real-time assessments of solitary experiences in the paragraph on page 27:

“A second limitation is that the daily diary method did not involve real-time assessments of solitary experiences. Instead, we measured participants’ retrospective evaluations of individuals’ preference and motivation for solitude at the end of each day. As such, it was not possible for us to investigate whether any situational or momentary factors that happen prior to solitude that could have contributed to participants’ preference and motivation for this time. For example, our data suggested that some individuals’ preference for solitude might vary depending on the day while others might prefer solitude more consistently across days. This means that, for some people, a moderator might explain why someone would prefer to spend time alone on a particular day, or that some kinds of events may enhance this preference. Research on solitude seeking after being ostracized (20,45) could provide some insights here. It may be that individuals with certain personality traits might be more likely to prefer solitude after feeling ostracized or excluded by others.”

13. The Methods and Results section headings should specify whether they’re referring to Study 1 or Study 2 (to avoid ambiguity)

Response. We have now done this. For example, Method now reads “Study 1 Method”

14. On p. 5: “In contrast, in another study that examined preference for solitude by Burger’s operationalization and self-determined motivation for solitude from SDT perspective in older adults showed positive correlations between these two concepts.” Which study is this sentence referring to? And in which age group? The implication in this paragraph is that mixed findings may be due to different motivations across age groups, but it is not clear exactly where this conclusion comes from.

Response. The sentence now reads: “Another study that examined preference for solitude in adults older than 35 years of age showed positive correlation between Burger’s preference for solitude scale and measure of self-determined motivation for solitude, yet negative correlation between preference and measure of not self-determined motivation for solitude (23)” to elaborate on the details of the study. We have also added the citation 23 on p. 5.

23. Weinstein N, Nguyen TT. Motivation and preference in isolation: A test of their different influences on responses to self-isolation during the COVID-19 outbreak. R Soc Open Sci. 2020;7(5).

15. In the study 1 Participants section (p. 9), it is stated “One hundred and eighty three undergraduate students (153 females; 9% Hispanic), and then it is stated, “The sample consisted of 53% Whites and Caucasians, 35% Asians or Pacific Islanders, 5% Black or African Americans, …” This double description of the sample ethnic make-up is a bit confusing as they do not match.

Response. Thank you, good point. We have removed this other statistic as it was not accurate.

In summary, we are grateful for the constructive feedback we received from the editor and both reviewers. Their views helped us to clarify, elaborate, and correct text in key places that improved how we communicated the conceptual approach of the project, methodological decisions made, and analytic decisions and findings. We find the revised manuscript is much improved and hope that readers see the same.

Sincerely, 

Thuy-vy T. Nguyen

Netta Weinstein

Richard M. Ryan

---

## [Decision Letter · Decision Letter 1]

20 Jan 2022

PONE-D-21-07216R1Who Enjoys Solitude? Autonomous Functioning (But Not Introversion) Predicts Self-Determined Motivation (But Not Preference) for SolitudePLOS ONE

Dear Dr. Nguyen,

Thank you for re-submitting your manuscript to PLOS ONE. After careful consideration, we still feel that the paper has merit but does not fully meet PLOS ONE’s publication criteria as it currently stands. Therefore, we invite you to submit a revised version of the manuscript that addresses the points raised during this round of the review process. The issues with the paper on this round are mostly mine, and reflect ongoing concerns with your sample size and population.

In the previous round, I asked: 1) How were sample sizes determined? Power analyses are needed here, or some other justification for the sample sizes. (2) Why were undergraduate students chosen as the population of interest (beyond convenience)?

In response to question 1, you responded that “We believe it is justified to determine sample sizes based on realistic measures: expectations of how many participants we can recruit from the pool and lack of external funding to recruit outside of this pool.”

This explanation is inadequate. Consider a study that drew from a very small pool of participants and had no funding support, permitting a recruitment of 10 people. Would this be sufficient justification for that sample size? I think not. If your studies are not appropriately powered the reader doesn't know how reliable the findings are. This casts your study findings into considerable doubt. The lack of power analyses for both studies, and the consequences thereof, should be clearly acknowledged in your General Discussion.

In response to my question 2, you stated that: “We selected to test study hypotheses with students because we could access and track this population; however, it is worth noting that this age group – representing emerging adulthood – have been the focus of past research which has found robust links between self-determined motivation and well-being in solitude...”

While I find your rationale sensible, it doesn't account for the fact that students may be a poorly generalizable population in which to address the study questions. I recommend that you acknowledge the limitations of college student sampling in your General Discussion.

Please also address the comments of Reviewer 1, who asks for clarification on several points.

While I am calling this a minor revision, your attention to these issues will determine the final decision on the manuscript.

We look forward to receiving your revised manuscript.

Kind regards,

Kirk Warren Brown

Academic Editor

PLOS ONE

Journal Requirements:

Reviewers' comments:

Reviewer's Responses to Questions

**Comments to the Author**

1. If the authors have adequately addressed your comments raised in a previous round of review and you feel that this manuscript is now acceptable for publication, you may indicate that here to bypass the “Comments to the Author” section, enter your conflict of interest statement in the “Confidential to Editor” section, and submit your "Accept" recommendation.

Reviewer #1: All comments have been addressed

Reviewer #2: All comments have been addressed

2. Is the manuscript technically sound, and do the data support the conclusions?

Reviewer #1: Yes

Reviewer #2: Yes

3. Has the statistical analysis been performed appropriately and rigorously? 

Reviewer #1: Yes

Reviewer #2: Yes

4. Have the authors made all data underlying the findings in their manuscript fully available?

Reviewer #1: Yes

Reviewer #2: Yes

5. Is the manuscript presented in an intelligible fashion and written in standard English?

Reviewer #1: Yes

Reviewer #2: Yes

6. Review Comments to the Author

Reviewer #1: Thank you to the authors for adequately addressing my comments and questions in the first round of reviews. Two minor comments/edits: 1) It's still not clear in the Procedure section of Study 1 what measures or questions were asked in the daily surveys. The initial survey is clearly explained, but I have to infer what was asked in the daily surveys. The statement on p. 14 under "Diary Survey" is a bit vague ("For each diary surveys [sic], participants reported the events that happened to them that day, including their experiences with social and solitary experiences.") and I had to assume that the measures for dispositional autonomy and introversion measures were included in each daily survey, but this was not explicitly stated. 2) Change AFI to IAF on p. 13.

Reviewer #2: Thank you for thoroughly addressing my comments in this revision - I have no further concerns. The manuscript was a pleasure to read and I think it makes a valuable contribution to our understanding of the role of autonomous functioning in motivation for solitude.

7. PLOS authors have the option to publish the peer review history of their article (what does this mean?). If published, this will include your full peer review and any attached files.

Reviewer #1: No

Reviewer #2: **Yes: **Jennifer Lay

---

## [Author Response · Author response to Decision Letter 1]

23 Mar 2022

Sample size justification

The Editor commented:

“We believe it is justified to determine sample sizes based on realistic measures: expectations of how many participants we can recruit from the pool and lack of external funding to recruit outside of this pool.” This explanation is inadequate. Consider a study that drew from a very small pool of participants and had no funding support, permitting a recruitment of 10 people. Would this be sufficient justification for that sample size? I think not. If your studies are not appropriately powered the reader doesn't know how reliable the findings are. This casts your study findings into considerable doubt. The lack of power analyses for both studies, and the consequences thereof, should be clearly acknowledged in your General Discussion.

Response: Since I did not run power analysis to determine our sample sizes for Study 1 and Study 2, I conducted power simulations to examine what is the smallest detectable effect size, given multiple possible sample sizes. I have now added this in the Discussion on page 24: “Since we did not conduct power analyses to determine our sample sizes for both studies, we conducted power calculation on simulated data. This is more appropriate than performing post-hoc power analyses, which have been criticized for not providing true observed power when being conducted on data that has been collected and analysed (47). In a series of power simulations (n = 100 simulations) looking at achieved power across sample sizes between 140 and 270 and assuming 5 to 7 data points per participant, a sample of 270 (Study 2’s sample) should allow 80% power to detect an effect size as small as .08.”

Reference: Zhang Y, Hedo R, Rivera A, Rull R, Richardson S, Tu XM. Post hoc power analysis: Is it an informative and meaningful analysis? Gen Psychiatry. 2019;32(4):3–6.

The Editor commented:

In response to my question 2, you stated that: “We selected to test study hypotheses with students because we could access and track this population; however, it is worth noting that this age group – representing emerging adulthood – have been the focus of past research which has found robust links between self-determined motivation and well-being in solitude...”

While I find your rationale sensible, it doesn't account for the fact that students may be a poorly generalizable population in which to address the study questions. I recommend that you acknowledge the limitations of college student sampling in your General Discussion.

Response: I have now strengthened this point in the General Discussion on page 28: “A third limitation is that both studies relied on college or university samples. In this research, we recruited participants within the typical age range of those who attend universities in the United States, with a mean age of 20 years, lending caution to any generalizations to the larger population beyond young adults who attend higher education in the US. Whether education levels and different life circumstances might contribute to motivation and preference for solitude is an empirical question that warrants future research.”

Finally, Reviewer 1 commented:

“1) It's still not clear in the Procedure section of Study 1 what measures or questions were asked in the daily surveys. The initial survey is clearly explained, but I have to infer what was asked in the daily surveys. The statement on p. 14 under "Diary Survey" is a bit vague ("For each diary surveys [sic], participants reported the events that happened to them that day, including their experiences with social and solitary experiences.") and I had to assume that the measures for dispositional autonomy and introversion measures were included in each daily survey, but this was not explicitly stated. 2) Change AFI to IAF on p. 13.” 

Response: I have added more details to the Procedure section of Study 1 on page 14: “For each diary survey, participants were asked about a significant event that happened to them that day, their experiences of such event (e.g., autonomy, need satisfaction, positive and negative affect), their self-determined motivation for solitude and levels of preference for solitude. Only self-determined motivation for solitude and preference for solitude were the variables of interest for this present paper, whereas other questions were part of a separate project studying the link between daily positive events and later memories of such events.” 

For the measures for dispositional autonomy and introversion measures, I believe the reviewer might have missed what I said on page 13: “All participants were enrolled after submitting an initial survey. Included in the initial survey were several personality measures, including the Introversion subscale from the Big-Five Inventory (40) and the Index of Autonomous Functioning (IAF) (27). Descriptions of personality measures used in this study are provided below.”

I hope those revisions are sufficient in addressing the comments. I am looking forward to any further feedback to help improve this manuscript. 

Sincerely,

Thuy-vy Nguyen

---

## [Editor Report · Decision Letter 2]

5 Apr 2022

Who Enjoys Solitude? Autonomous Functioning (But Not Introversion) Predicts Self-Determined Motivation (But Not Preference) for Solitude

PONE-D-21-07216R2

Dear Dr. Nguyen,

We’re pleased to inform you that your manuscript has been judged scientifically suitable for publication and will be formally accepted for publication once it meets all outstanding technical requirements.

Kind regards,

Kirk Warren Brown

Academic Editor

PLOS ONE
---

## [Editor Report · Acceptance letter]

4 May 2022

PONE-D-21-07216R2 

Who Enjoys Solitude? Autonomous Functioning (But Not Introversion) Predicts Self-Determined Motivation (But Not Preference) for Solitude 

Dear Dr. Nguyen:

I'm pleased to inform you that your manuscript has been deemed suitable for publication in PLOS ONE. Congratulations! Your manuscript is now with our production department. 

Kind regards, 

on behalf of

Dr. Kirk Warren Brown 

Academic Editor

PLOS ONE